# A genetic screen identifies new steps in oocyte maturation that enhance proteostasis in the immortal germ lineage

Madhuja Samaddar[1†], Jérôme Goudeau[1†], Melissa Sanchez[2], David H Hall[3], K Adam Bohnert[1‡], Maria Ingaramo[1], Cynthia Kenyon[1]*

[1]Calico Life Sciences LLC, South San Francisco, United States; [2]Department of Molecular and Cellular Biology, University of California, Berkeley, Berkeley, United States; [3]Dominick P. Purpura Department of Neuroscience, Albert Einstein College of Medicine, New York, United States

**Abstract** Somatic cells age and die, but the germ-cell lineage is immortal. In *Caenorhabditis elegans*, germline immortality involves proteostasis renewal at the beginning of each new generation, when oocyte maturation signals from sperm trigger the clearance of carbonylated proteins and protein aggregates. Here, we explore the cell biology of this proteostasis renewal in the context of a whole-genome RNAi screen. Oocyte maturation signals are known to trigger protein-aggregate removal via lysosome acidification. Our findings suggest that lysosomes are acidified as a consequence of changes in endoplasmic reticulum activity that permit assembly of the lysosomal V-ATPase, which in turn allows lysosomes to clear the aggregates via microautophagy. We define two functions for mitochondria, both of which appear to be independent of ATP generation. Many genes from the screen also regulate lysosome acidification and age-dependent protein aggregation in the soma, suggesting a fundamental mechanistic link between proteostasis renewal in the germline and somatic longevity.

*For correspondence:
cynthia@calicolabs.com

[†]These authors contributed
equally to this work

Present address: [‡]Department
of Biological Sciences, Louisiana
State University, Baton Rouge,
United States

Competing interest: See
page 27

Reviewing editor: Jan Gruber,
Yale-NUS College, Singapore

## Introduction

The germ cells of an organism not only give rise to the rich diversity of somatic lineages, they also generate new germ cells that, by seeding subsequent generations, confer species immortality. This sequential rejuvenation of youthfulness is one of the most interesting mysteries in biology. How is it achieved? Not surprisingly, a process so crucial to the survival of the species involves multiple levels of quality control. At one level, an entire oocyte can be eliminated via apoptosis; for example, following DNA damage (*Hunter, 2017*; *Luo et al., 2010*; *Qiao et al., 2018*; *Schumacher et al., 2001*). Oocytes can also eliminate defective organelles or age-related molecular damage. For example, in mice, mitochondria containing defective DNA are eliminated during oogenesis (*Fan et al., 2008*), and in yeast, nuclear senescence factors and protein aggregates are extruded during meiosis (*King et al., 2019*; *Unal et al., 2011*). Mechanisms likely exist to counteract protein aggregation in *Drosophila* as well: relative to the soma, the eggs of *Drosophila* exhibit enhanced proteasome activity and resistance to protein aggregation (*Fredriksson et al., 2012*). Autophagy, a process known to eliminate damaged organelles and macromolecules, may play a role as well as it is required for the survival of the mouse germ lineage (*Gawriluk et al., 2011*).

The nematode *C. elegans* is a valuable organism for studies of germline quality control. The animal is transparent and genetically accessible, allowing one to analyze the effects of perturbations over time in living animals using fluorescent reporters. *C. elegans* is hermaphroditic, producing both sperm and oocytes in each of two symmetrical gonadal arms. Germ cells originate in stem-cell niches located at the distal ends of each arm and pass through successive stages of meiosis as they move

proximally towards the spermatheca, where fertilization occurs (*Crittenden et al., 1994*; *Hubbard and Greenstein, 2005*; *McCarter et al., 1999*; *Figure 1A*). In 2010, Goudeau and Aguilaniu discovered that carbonylated proteins, known to accumulate during aging, are present throughout the distal germline, but are eliminated in proximal oocytes (*Goudeau and Aguilaniu, 2010*). Mutant worms that lack sperm and develop as females retain carbonylated proteins throughout the entire germline, indicating that sperm play a role in their elimination. In addition to carbonylated proteins, protein aggregates also accumulate in the proximal oocytes of females but not in those of hermaphrodites (*Bohnert and Kenyon, 2017*). Signals from sperm trigger the elimination of protein aggregates in a process that is coupled to the broader choreography of oocyte maturation, which involves dramatic changes in organelle morphology and function (*Bluemink et al., 1983*; *Carroll, 1996*; *Charbonneau and Grey, 1984*; *FitzHarris et al., 2007*; *Kobayashi et al., 1991*; *Maller et al., 1977*; *Mehlmann et al., 1995*; *Wasserman et al., 1982*; *Yamashita, 2018*). In *C. elegans,* oocyte maturation is triggered by actin-like major sperm proteins (MSPs) released from sperm (*Kosinski et al., 2005*; *Miller et al., 2003*; *Miller et al., 2001*) and involves changes in chromosome and nuclear dynamics linked to progression through meiosis, upregulation of translation, and changes in endoplasmic reticulum (ER), lysosomal and mitochondrial function (*Bohnert and Kenyon, 2017*; *Huelgas-Morales and Greenstein, 2018*; *Langerak et al., 2019*). The shift in lysosomal physiology that occurs during oocyte maturation is crucial for aggregate removal: the sperm-derived signals trigger the acidification of lysosomes, which in turn engulf the aggregates by a process that resembles microautophagy morphologically (*Bohnert and Kenyon, 2017*). Lysosomal acidification also triggers a conversion in mitochondrial dynamics from what appears to be a poised state, characterized by a high membrane potential, to an active state, in which the membrane potential is reduced. Conditions that prevent this mitochondrial membrane potential shift, such as inactivation of mitochondrial ATP synthase, prevent aggregate removal. However, how lysosomes become acidified in response to sperm signals, how mitochondria contribute to aggregate removal, and whether other organelles also play a role in aggregate clearance are important outstanding questions.

To better understand mechanisms of proteostasis enhancement in the *C. elegans* germ lineage, we combined cell biological approaches with a confocal microscopy-based genome-wide RNAi screen for gene knockdowns that cause protein aggregates to accumulate even in the presence of sperm. Our findings deepen our understanding of many aspects of protein aggregation and clearance within the germline. Specifically, we find that (i) protein aggregates accumulate in young, newly formed oocytes in the absence of sperm—their formation is an integral part of oocyte development rather than a consequence of prolonged oocyte quiescence; (ii) inhibiting processes required for changes in ER morphogenesis and function during oocyte maturation, including protein synthesis and actin dynamics, prevents lysosome acidification by preventing the assembly of the V-ATPase lysosomal proton pump; (iii) mitochondria elaborate an energy-independent checkpoint that gates protein-aggregate removal; (iv) endosomal sorting complex required for transport (ESCRT) proteins are required for aggregate removal, consistent with protein aggregates being removed via microautophagy; and, unexpectedly (v) as with proteasome inhibition, inhibiting TRiC-complex chaperonins or HSP70 chaperones promotes protein aggregation indirectly by acting upstream in the pathway to block lysosome acidification. Finally, we show that inhibiting lysosomal V-ATPase, vesicle-transport or proteasome function also accelerates age-dependent protein aggregation in the soma, thus bridging mechanisms that enhance proteostasis in the immortal germ lineage to those that maintain proteostasis in the aging soma.

## Results

### A genome-wide RNAi screen identifies 81 genes that influence oocyte protein aggregation

To undertake an unbiased genome-wide RNAi screen for components of the germline proteostasis network, we utilized the Ahringer RNAi feeding library (*Kamath et al., 2003*), which targets ~87% of annotated *C. elegans* genes, to identify gene inhibitions that cause proteins to aggregate even in the presence of sperm; that is, in hermaphrodites. A germline-specific GFP::RHO-1 translational fusion was selected as the reporter for the screen as it forms dense patchy aggregates in female oocytes (*Bohnert and Kenyon, 2017*) and in unfertilized oocytes of aging hermaphrodites, which

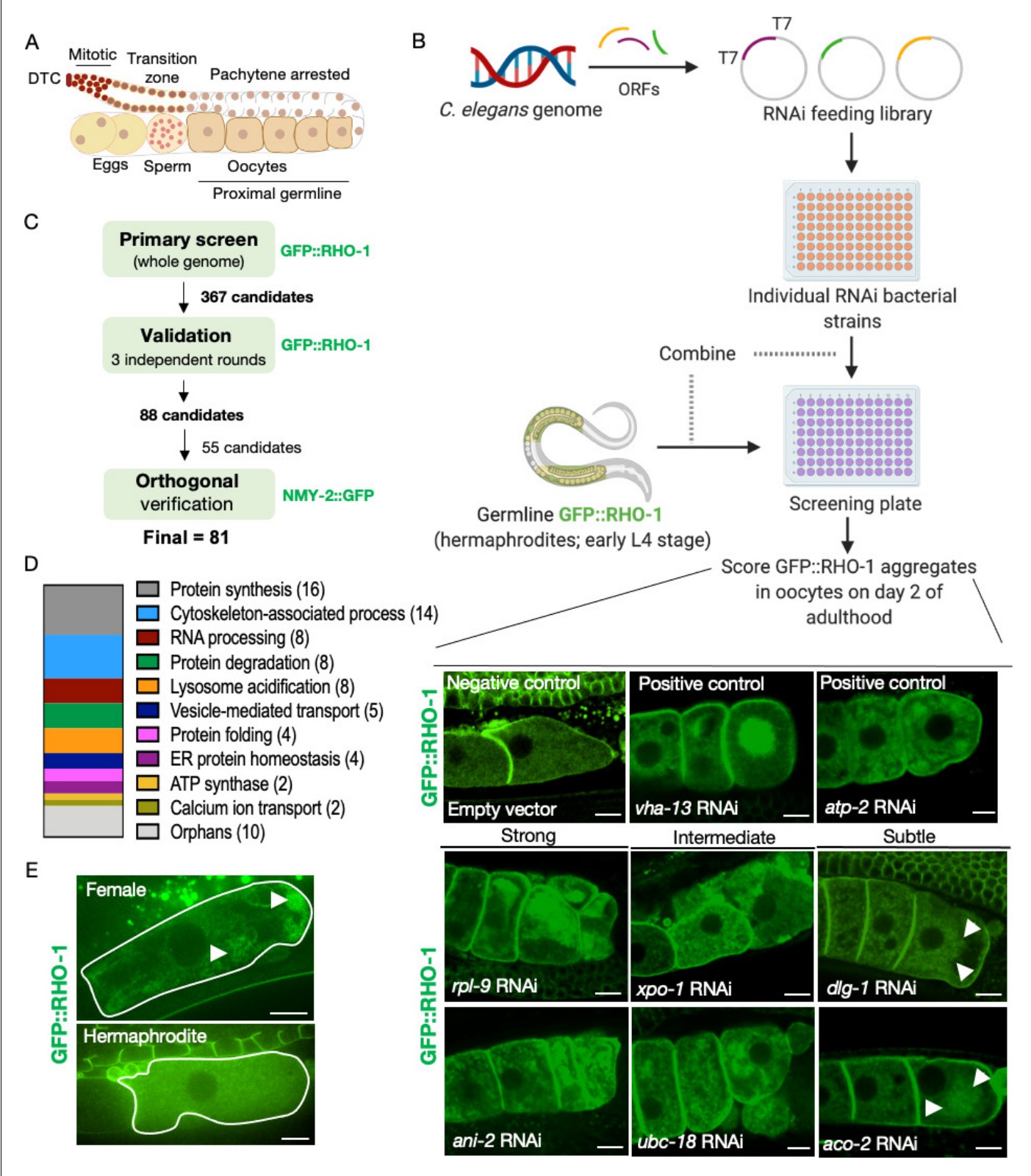

**Figure 1.** Identification of genes that prevent protein aggregation in maturing *C. elegans* oocytes. (**A**) Diagram of the *C. elegans* germline. (**B**) Genome-wide RNAi-screen workflow. High-resolution images from the validation rounds are represented here. Scale bars, 10 μm; white arrows, GFP:: RHO-1 aggregates in subtle hits. (**C**) Schematic representation of the procedure used for candidate validation and orthogonal verification using two independent germline-specific reporters. 45 candidates representing the major gene ontology (GO) categories, and all 10 orphans were subjected to

*Figure 1 continued on next page*

*Figure 1 continued*

orthogonal verification. (D) Common biological process GO categories enriched in the screen hits. Genes that were not members of these enriched categories were classified as orphans. (E) Very early oocytes of females and hermaphrodites (CF4557 and CF4552 strains, respectively). GFP::RHO-1 aggregates often appear in newly formed female oocytes. Scale bars, 10 µm; white arrows, GFP::RHO-1 aggregates.

The online version of this article includes the following figure supplement(s) for figure 1:

**Figure supplement 1.** Additional examples of phenotypes from the primary and the orthogonal screens, and the most-highly enriched GO-terms.

**Figure supplement 2.** Visualizing ER morphology and GFP::RHO-1 aggregates in very early oocytes.

deplete their stores of sperm (*David et al., 2010*). Oocytes are formed in young adults, so RNAi was initiated at the beginning of the last larval stage (L4) to reduce the chance of missing genes with essential roles during development (*Figure 1B*). The animals were examined manually by confocal microscopy on day 2 of adulthood for the presence of GFP::RHO-1 aggregates in proximal oocytes (*Figure 1A, B*, *Figure 1—figure supplement 1A*). The primary screen identified 367 genes whose knockdown led to GFP::RHO-1 aggregation in at least 50% of the animals. To eliminate false positives, the candidates were subjected to three independent rounds of validation using a similar workflow, with at least two independent rounds from two experimentalists (*Figure 1C*). We note that false negatives could result from failure of an RNAi bacterial strain to grow adequately, from insufficient or excessive gene inhibition, or because of functional redundancy.

The resulting set of 88 genes represented diverse cellular processes (*Supplementary file 1*), with highly enriched gene ontology (GO) categories including translation, RNA processing, and cytoskeletal organization (*Figure 1—figure supplement 1C*). In addition, ATP-hydrolysis-coupled proton transport and lysosome acidification were both highly enriched, in agreement with our previous findings (*Bohnert and Kenyon, 2017*). We sorted the candidates according to relative representation in major biological processes (*Figure 1D*), after discarding candidates with known roles in male gonad and germ cell development (*fkh-6, spe-15*), in regulation of our proteostasis marker RHO-1 (*cyk-4*), or with potential off-target effects (*adr-1* and *dhs-5*) (*Supplementary file 1*). Protein synthesis and degradation, RNA processing, protein folding, and lysosome acidification were strongly represented biological processes, as were genes involved in cytoskeletal organization, including four of the five *C. elegans* actin genes. Among organelle systems, multiple candidates were associated with ER homeostasis (*hsp-4, spcs-1,* and *srpa-68*) and vesicle-mediated transport (*copb-2, sec-24.1, sar-1, rab-11.1,* and *E03H4.8*). We also identified genes that could not be classified into any of the highly enriched biological process categories and were designated 'orphans' (*Supplementary file 1* and *Figure 1D*).

We subjected representative candidates from each group, as well as all of the orphans, to an orthogonal screening approach, this time utilizing a different aggregation reporter, a germline-specific NMY-2::GFP fusion protein (*Bohnert and Kenyon, 2017*) that forms distinctive punctate aggregates in the absence of sperm (*Figure 1C* and *Supplementary file 1*). Out of the 55 candidates tested, 53 RNAi knockdowns also caused NMY-2::GFP protein-aggregate accumulation (*Figure 1—figure supplement 1B*). Thus, these genes are likely to have a general effect on protein-aggregate clearance rather than a specific effect on GFP::RHO-1.

## Aggregates form in young proximal oocytes

We classified the RNAi phenotypes as strong, intermediate or subtle based on the degree of GFP::RHO-1 aggregation (*Supplementary file 1*). Typically, the phenotypically strong gene knockdowns led to sterility, whereas the subtle knockdowns generally did not, though some reduced brood size. In principle, one could imagine that aggregates accumulate in oocytes from sterile hermaphrodites because oocytes that would normally undergo ovulation and pass into the uterus do not, and instead simply 'age in place.' Likewise, aggregates could accumulate in normal female oocytes because in the absence of sperm they proceed much more slowly through the spermatheca and into the uterus (*McCarter et al., 1999*). As a consequence, proximal oocytes of females are technically older than those of age-matched hermaphrodites (*Kim et al., 2013*). This oocyte-age discrepancy raises the question of whether aggregates accumulate only during long-term oocyte quiescence or whether they are present in newly formed female oocytes. To address this question, we collected age-matched late L4 ('Christmas-tree' stage) females and hermaphrodites and followed the

hermaphrodites until their first oocytes were cellularized. We found that age-matched first-proximal oocytes of young females often contained protein aggregates (*Figure 1E, Figure 1—figure supplement 2*). Therefore, protein-aggregate accumulation does not require prolonged oocyte quiescence; instead, it appears to be part of the normal developmental sequence of oocytes that are not exposed to sperm.

## The ER influences protein aggregation

Many genes we identified were associated with the ER, so we explored ER biology in more detail. The ER undergoes a major morphological rearrangement in response to oocyte maturation signals (*Langerak et al., 2019*). We visualized the ER in *C. elegans* oocytes using a germline-specific mCherry::SP12 reporter (*Joseph-Strauss et al., 2012*). SP12 is a signal-peptidase subunit (*Poteryaev et al., 2005*), and *spcs-1*, which encodes SP12, was identified in our screen. In hermaphrodite oocytes, the ER assumed a fine network appearance, whereas in female oocytes, the ER formed bright, distinctive patches (*Figure 2A, C*, *Figure 2—figure supplement 1A*). We also examined ER morphology in the first, newly formed female oocytes using the L4 staging procedure described above. Pristine female oocytes also exhibited ER patches, indicating that the altered ER architecture is an intrinsic property of female oocytes and not a consequence of prolonged quiescence (*Figure 2B, Figure 2—figure supplement 1B*). The fluorescent signal from the female ER was further enhanced as oocytes aged, possibly due to continued ER biogenesis coupled with oocyte compression caused by their packing within the gonad (*Figure 2—figure supplement 1A*). Using two fluorescent reporters, we visualized both the ER and GFP::RHO-1 simultaneously (*Figure 1—figure supplement 2*, *Figure 2B*, *Figure 2—figure supplement 1C*, *Figure 2—figure supplement 2*) and found that in female oocytes the ER and GFP::RHO-1 aggregates segregated spatially into mutually exclusive regions (*Figure 1—figure supplement 2*, *Figure 2B*, *Figure 2—figure supplement 1C*).

We also examined oocytes from young, age-matched hermaphrodites and females by transmission electron microscopy (TEM). The ER in pristine oocytes from both sexes appeared to be associated with ribosomes, indicative of active (or poised) protein synthesis, yet the female ER was distinctively clustered into parallel stacks (*Figure 2D, Figure 2—figure supplement 3*).

To test whether the stacked ER architecture of female oocytes could be reversed by sperm, we visualized fluorescently tagged ER and NMY-2::GFP aggregates simultaneously over time in female oocytes after mating. The presence of sperm triggered distinctive remodeling of the ER architecture and the clearance of aggregates, although with different kinetics (*Figure 2E, Figure 2—figure supplement 4*). The earliest changes in the ER were detected approximately 10 min after mating, and the morphology was significantly altered by 30 min (*Figure 2E*). In contrast, the clearance of NMY-2 aggregates started a few minutes later, with aggregate clearance in the most proximal oocytes typically occurring between 20 and 30 min after mating (*Figure 2—figure supplement 4, 5, 6A*).

Next, we investigated whether the genes identified in our RNAi screen might contribute to proteostasis by influencing a process associated with ER architecture. To this end, we subjected mCherry::SP12-germline-labeled hermaphrodites to 35 RNAi treatments representing the various biological process categories as well as the individual orphans. This subset of RNAi clones, which we termed the 'assay pool,' was used for all subsequent large-scale phenotypic analyses (*Table 1*). We visualized oocyte ER morphology in young adults subjected to the gene knockdowns (*Figure 2—figure supplement 6B* and *Table 1*). Genes involved in cytoskeletal organization, protein synthesis and protein folding were required for maintaining normal ER morphology and led to an altered ER architecture when knocked down. Genes involved in RNA processing, protein degradation and trafficking comprised a mixed class, with some knockdowns appearing to affect ER morphology. We note that our analysis does not distinguish between genes with direct effects on ER morphogenesis, like the actin cytoskeleton (*Poteryaev et al., 2005*), and genes with indirect effects, like the proteasome, which is required for the degradation of GLD-1 (*Bohnert and Kenyon, 2017*; *Goudeau et al., 2020*; *Spike et al., 2018*), a translational repressor involved in oocyte cell fate determination (*Francis et al., 1995*; *Jones et al., 1996*). Notably, disrupting the mitochondrial ATP synthase, the V-ATPase complex, or ER function itself did not visibly alter the ER network, suggesting that these gene functions were not required for the sperm-induced ER morphology change, or more generally for ER morphogenesis.

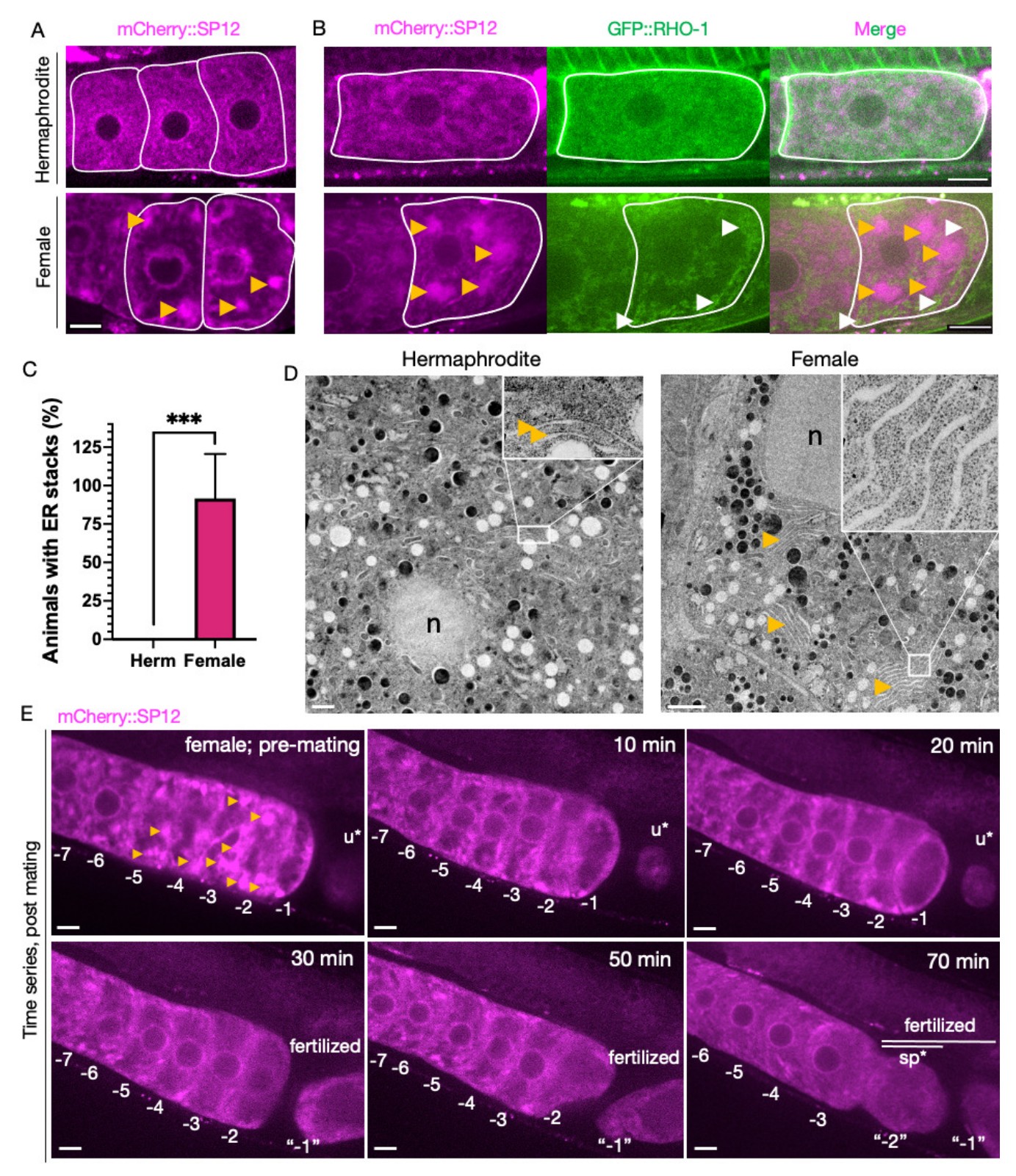

**Figure 2.** Endoplasmic reticulum (ER) morphology is regulated by sperm and correlates with protein aggregation. (**A**) ER morphology in the germline of hermaphrodite (OCF15) and female (CF4542) animals, visualized using an mCherry::SP12 reporter. Orange arrows, ER patches. Scale bar, 10 μm. (**B**) GFP::RHO-1 localization in newly formed oocytes from hermaphrodites (CF4552) and females (CF4557) also expressing mCherry::SP12. White arrows, GFP::RHO-1 aggregates. Scale bars, 10 μm. (**C**) Percentage of animals with ER patches in oocytes. n = 12; mean ± sd, ***p≤0.001 (Mann–Whitney test).

*Figure 2 continued on next page*

*Figure 2 continued*

(D) Transmission electron micrographs (TEM) from hermaphrodite (N2E) and female (CF4101) oocytes. At least eight females and eight hermaphrodites were subjected to TEM. Orange arrows, ER. Left panel, magnification ×1400, scale bar, 1 μm; inset: magnification ×13,000, scale bar, 100 nm. Right panel, magnification ×1200, scale bar, 2 μm; inset: magnification ×18,500, scale bar, 100 nm. (E) Dissolution of ER patches (orange arrows) in female oocytes upon mating with CB1490 males. Female oocytes were imaged before and after mating at the indicated time points. Each oocyte is numbered according to its initial position relative to the spermatheca, and original numbers are maintained after fertilization. The figure shows a single representative animal from seven independent experiments. u*, unfertilized oocyte/debris in uterus in a different focal plane; sp*, an oocyte passing through the spermatheca during fertilization. Scale bars, 10 μm.

The online version of this article includes the following figure supplement(s) for figure 2:

**Figure supplement 1.** Endoplasmic reticulum (ER) morphologies in hermaphrodite and female oocytes.
**Figure supplement 2.** Supplementary data related to *Figure 2B*.
**Figure supplement 3.** Supplementary data related to *Figure 2D*.
**Figure supplement 4.** Simultaneous visualization of the alteration in endoplasmic reticulum (ER) morphology and clearance of aggregates in female oocytes (CF4560) as a function of time, following mating.
**Figure supplement 5.** Simultaneous visualization of oocyte ER morphology and protein aggregates in an unmated female (CF4560).
**Figure supplement 6.** Changes in NMY-2::GFP signal following mating, as a function of time; and representative ER morphologies in assay-pool knockdowns.

## Sperm signals initiate V-ATPase assembly, enabling lysosomal acidification

Signals from sperm elicit the lysosomal acidification required for the clearance of protein aggregates, and inhibiting the V-ATPase, which acidifies lysosomes, causes proteins to aggregate in hermaphrodite oocytes (*Bohnert and Kenyon, 2017*). However, the mechanism by which lysosome acidification occurs in response to signals from sperm is not known (*Goudeau et al., 2020*; *Spike et al., 2018*). To address this question with minimal physiological disruption, we first attempted to fluorescently tag endogenously-expressed V-ATPase subunits using CRISPR/Cas9. We picked four different subunits, two from the membrane-embedded $V_0$ domain (VHA-4 and VHA-7) and two from the peripheral $V_1$ domain (VHA-11 and VHA-13) (*Supplementary file 2*). Unexpectedly, although each one of these tagged proteins was visible in somatic tissues, we were unable to detect any fluorescence in the germline in either intact animals (*Figure 3—figure supplement 1*) or in their dissected gonads (*Figure 3—figure supplement 2A*). Germline fluorescence was also undetectable when imaging with a more sensitive wide-field system or following photobleaching of the strong intestinal signal, which we reasoned might mask a faint signal from the germline (data not shown). Endogenous genes tagged with fluorescent proteins can be silenced in the germline by the process of epigenetic RNAe (*Shirayama et al., 2012*); however, this explanation seems unlikely since, unlike V-ATPase RNAi-treated animals, these animals produced viable embryos.

We also considered the possibility that these V-ATPase proteins did not act cell-autonomously in the gonad, though V-ATPase mRNAs previously had been identified as direct targets of the germline-specific GLD-1 translational regulator (*Wright et al., 2011*). To investigate cell autonomy, we carried out tissue-specific RNAi experiments using sterility as a phenotypic readout. Three hermaphrodite strains (wild type), globally RNAi-defective *rde-1(mkc36)* mutants DCL565, and germline-only RNAi-sensitive DCL569 animals (*Zou et al., 2019*) were subjected to RNAi from hatching (empty-vector, *vha-12* and *vha-13*). Both wild-type and germline-RNAi-permissive DCL569 animals exhibited sterility when fed *vha-12* and *vha-13* RNAi bacteria (*Figure 3—figure supplement 2B*), whereas the whole-body RNAi-defective *rde-1* mutant did not. Thus, these V-ATPase subunits are expressed and required in the germline, despite our inability to detect them when tagged endogenously.

As a next-best approach for visualizing germline V-ATPase subunits, we obtained a Mos1-mediated single copy insertion (MosSCI) strain of GFP::VHA-13 driven by the germline-specific *pie-1* promoter. This strain expressed GFP::VHA-13 in a genetically stable fashion, though presumably at elevated levels. In these animals, we observed a striking, sperm-dependent pattern: GFP::VHA-13 formed distinct foci in hermaphrodite oocytes (*Figure 3A*, *Figure 3—figure supplement 3A*) that colocalized with foci of LysoTracker, a dye that stains acidic lysosomes (*Figure 3B, Figure 3—figure supplement 4A*). In contrast, the proximal oocytes of females exhibited a diffuse GFP::VHA-13 fluorescent signal (*Figure 3A, Figure 3—figure supplement 3B*), with a slight enrichment in the perinuclear region. Female oocytes demonstrated very weak LysoTracker staining (*Bohnert and Kenyon,*

**Table 1.** Functional gene categories from the proteostasis screen.

| Biological process | RNAi target | Gene code | Normal ER architecture | GFP::VHA-13 puncta | Lysosome acidification | Reduction in ΔΨ |
|---|---|---|---|---|---|---|
| Protein degradation | pbs-7 | F39H11.5 | No | No | No | No |
| | rpn-6.1 | F57B9.10 | Mixed | No | No | No |
| Protein synthesis (translation) | rps-20 | Y105E8A.16 | No | No | No | No |
| | rpl-3 | F13B10.2 | No | No | No | No |
| Protein folding (chaperones) | cct-1 | T05C12.7 | No | Reduced | No | No |
| | cct-5 | C07G2.3 | No | Reduced | No | No |
| Cytoskeleton-associated process | act-1 | T04C12.6 | No | No | No | No |
| | pat-3 | ZK1058.2 | No | No | No | No |
| | ani-2 | K10B2.5 | No | No | No | No |
| ER protein homeostasis | hsp-4 | F43E2.8 | Yes | No | No | No |
| | spcs-1 | C34B2.10 | Yes | No | No | No |
| | srpa-68 | F55C5.8 | Yes | No | No | No |
| Vesicle-mediated transport (trafficking) | copb-2 | F38E11.5 | No | No | No | No |
| | sec-24.1 | F12F6.6 | Inconclusive | No | No | No |
| | sar-1 | ZK180.4 | Mixed | No | No | No |
| Lysosome acidification (V-ATPase) | vha-13 | Y49A3A.2 | Yes | No signal | No | No |
| | vha-12 | F20B6.2 | Yes | No | No | No |
| | vha-2 | R10E11.2 | Yes | No | No | No |
| ATP synthase | atp-3 | F27C1.7 | Yes | Yes | Yes | No |
| | atp-2 | C34E10.6 | Yes | Yes | Yes | No |
| RNA processing | ess-2 | F42H10.7 | No | Reduced | No | No |
| | cgh-1 | C07H6.5 | Mixed | No | No | No |
| | let-711 | F57B9.2 | Mixed | Reduced | No | No |
| Calcium ion transport | itr-1 | F33D4.2 | No | Reduced | No | No |
| | sca-1 | K11D9.2 | Yes | Reduced | No | No |
| Orphans | kin-2 | R07E4.6 | Mixed | No | No | No |
| | ttr-14 | T05A10.3 | Mixed | No | No | No |
| | par-5 | M117.2 | Yes | No | No | No |
| | xpo-1 | ZK742.1 | Yes | No | No | No |
| | srab-17 | T11A5.2 | Yes | Reduced | No | No |
| | rmd-2 | C27H6.4 | Yes | No | No | No |
| | dlg-1 | C25F6.2a.1 | No | No | No | No |
| | lgc-46 | Y71D11A.5 | Yes | Reduced | No | No |
| | aco-2 | F54H12.1 | Yes | No | No | No |
| | imb-1 | F28B3.8 | Inconclusive | No | No | No |
| ESCRT subunits* | vps-20 | Y65B4A.3 | Not tested | Yes | Yes | Not tested |
| | vps-28 | Y87G2A.10 | Not tested | Yes | Yes | Not tested |
| | vps-37 | CD4.4 | Not tested | Yes | Yes | Not tested |
| | vps-54 | T21C9.2 | Not tested | Yes | Yes | Not tested |

List of representative genes from each functional category that were selected for the assay pool and their effects on ER morphology, lysosome acidification, GFP::VHA-13 localization, and mitochondrial membrane potential.
*The ESCRT complex subunits were not derived from the original screen but from subsequent candidate testing.
ER: endoplasmic reticulum.

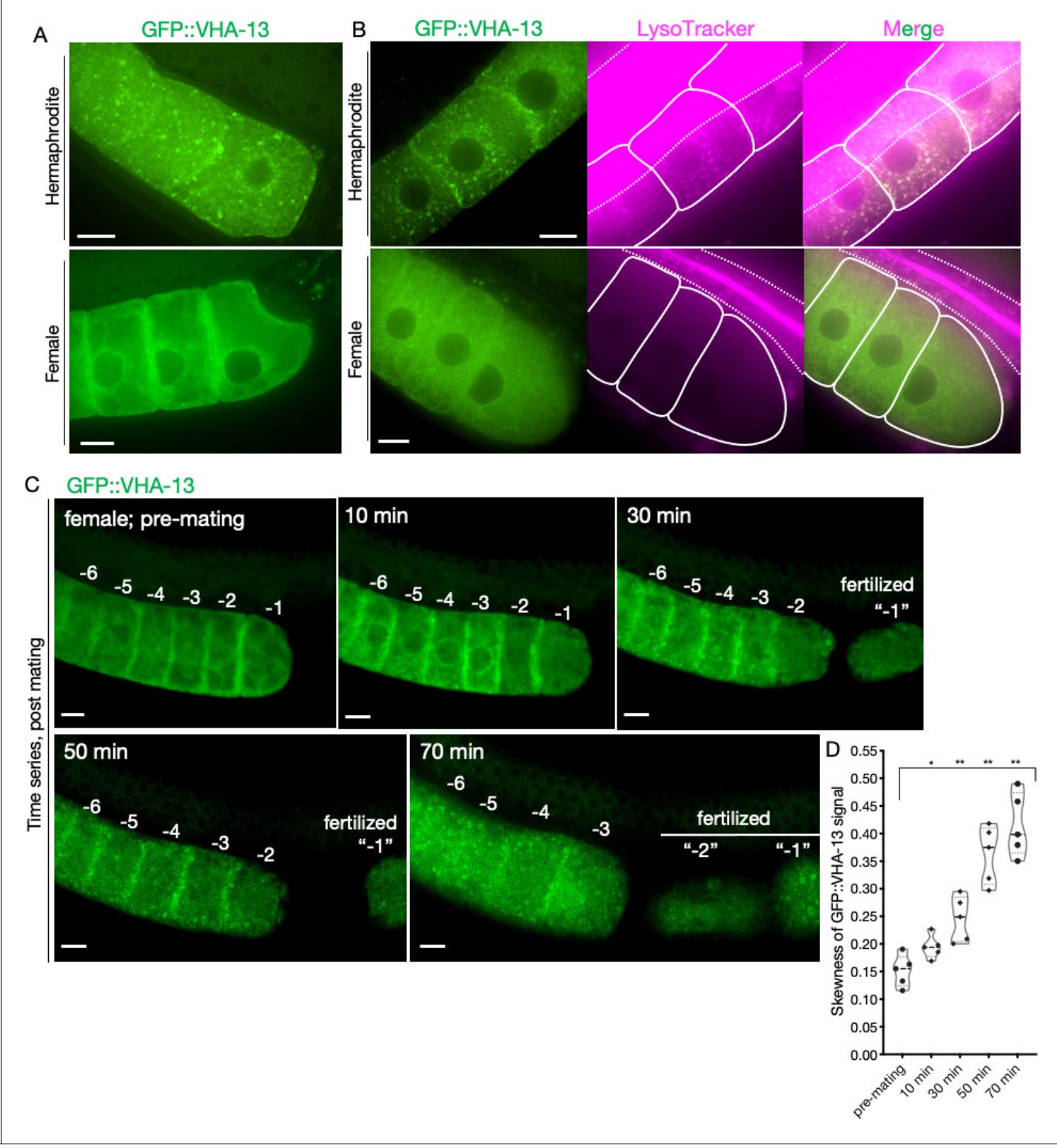

**Figure 3.** Sperm-dependent lysosomal localization of VHA-13. (A) Distribution of GFP::VHA-13 in proximal oocytes of hermaphrodites (PHX1414) and females (CF4599). (B) LysoTracker-stained oocytes from GFP::VHA-13-expressing hermaphrodites and females for visualization of acidic lysosomes. The two signals colocalized in hermaphrodite oocytes. Dotted line, intestinal boundary; solid line, oocytes. (C) Sperm signal-induced GFP::VHA-13 redistribution in female oocytes (CF4599). Oocytes were imaged before and after mating with CB1490 males. The GFP::VHA-13 signal shifts from diffuse to punctate following mating. Each oocyte is numbered according to its initial position relative to the spermatheca, and original numbers are maintained after fertilization. A representative individual is shown from five independent mating experiments. (D) Increase in signal skewness

*Figure 3 continued on next page*

*Figure 3 continued*

(inhomogeneity) of GFP::VHA-13 as a function of time following mating. Mann–Whitney test was used to determine statistical significance; *p≤0.05, **p≤0.01. Scale bars, 10 μm (**A–C**).

The online version of this article includes the following source data and figure supplement(s) for figure 3:

**Source data 1.** Skewness of oocyte GFP::VHA-13 signal as a function of time post mating.
**Figure supplement 1.** Attempts to visualize endogenously tagged V-ATPase subunits in the germline.
**Figure supplement 2.** Investigating VHA-13 expression and function in the germline.
**Figure supplement 3.** GFP::VHA-13 localization in the presence and absence of sperm (Supplementary data related to *Figure 3A*).
**Figure supplement 4.** Supplementary data related to *Figure 3B*.

*2017*), and there was no significant overlap in the two signals (*Figure 3B, Figure 3—figure supplement 4B*). To better control for effects of overexpression, we lowered the level of GFP::VHA-13 expression by serially diluting *gfp* RNAi bacteria. Even at the limit of fluorescence detection, hermaphrodites still exhibited GFP::VHA-13 foci, and females exhibited a diffuse distribution (*Figure 3—figure supplement 2C*). Thus, the germline harbors a mechanism by which VHA-13 is localized to discrete, acidic cellular locations, indicative of lysosomes, only in the presence of sperm. Notably, knocking down the *vha-12* gene, which encodes another peripheral V-ATPase $V_1$-domain subunit, also prevented GFP::VHA-13 foci formation in hermaphrodite oocytes (*Figure 3—figure supplement 2D*), suggesting a coordinated assembly pathway.

We asked whether sperm signals regulate V-ATPase localization in two ways. First, we examined older, sperm-depleted hermaphrodites (day 5) and observed diffuse V-ATPase localization, similar to that of females (*Figure 3—figure supplement 3C*). Second, we mated GFP::VHA-13-expressing females with males and monitored their proximal oocytes. The diffuse fluorescent signal began to change approximately 10 min after mating, organizing into distinct foci within 20–30 min (*Figure 3C, D*). This change coincided with release from arrest and oocyte maturation, ultimately leading to ovulation and fertilization. The time scales of ER remodeling and GFP::VHA-13 relocalization (*Figure 2E*) were strikingly similar, suggesting a possible causal relationship. Together, these findings suggest that sperm signals lead to lysosome acidification by initiating the recruitment of V-ATPase subunits to the surface of the lysosome and enabling proton-pump assembly.

## Most of the genes identified disrupt VHA-13 localization and lysosome acidification

To ask which gene inhibitions blocked lysosome acidification, we subjected LysoTracker-stained hermaphrodites to knockdowns of the 'assay pool' genes (*Table 1*). In wild-type hermaphrodites, lysosomes are acidified only in proximal oocytes (*Bohnert and Kenyon, 2017*). Therefore, to internally control for variations in dye uptake, we analyzed the ratios of staining intensities between proximal oocytes and distal germline regions. LysoTracker staining of hermaphrodites, females, and animals subjected to empty-vector RNAi were used as controls, and the simultaneous presence of GFP::RHO-1 aggregates confirmed successful gene knockdown. Unexpectedly, we found that the great majority of gene inhibitions prevented lysosome acidification (*Figure 4A, Figure 4—figure supplement 1A*). The ATP synthase genes *atp-2* and *atp-3* were the only exceptions, consistent with previous observations suggesting that mitochondrial ATP-synthase activity acts downstream of lysosome acidification to prevent protein aggregation (*Bohnert and Kenyon, 2017*).

Next, we asked whether reduced lysosomal acidification in the RNAi-treated hermaphrodites might be caused by an inability to localize the V-ATPase subunits to lysosomes, as occurs in females. To test this, we subjected GFP::VHA-13-expressing animals to assay pool RNAi and screened for the presence of GFP::VHA-13 puncta (*Figure 4—figure supplement 1C*). The extent of puncta formation was quantified by measuring the skewness (non-homogeneity) of the GFP::VHA-13 signal at multiple locations within the most proximal oocyte. Hermaphrodites and the empty-vector RNAi-treated animals demonstrated significantly higher skewness values than did female oocytes, which exhibited diffuse GFP::VHA-13 localization (*Figure 4B*). All of the knockdowns that blocked lysosome acidification showed reduced GFP::VHA-13 skewness, albeit to varying degrees. While most knockdowns resulted in a marked reduction in GFP::VHA-13 skewness, similar to the female, knockdown in some genes involved in protein folding (*cct-1, cct-5*), RNA processing (*ess-2, let-711*), and calcium ion

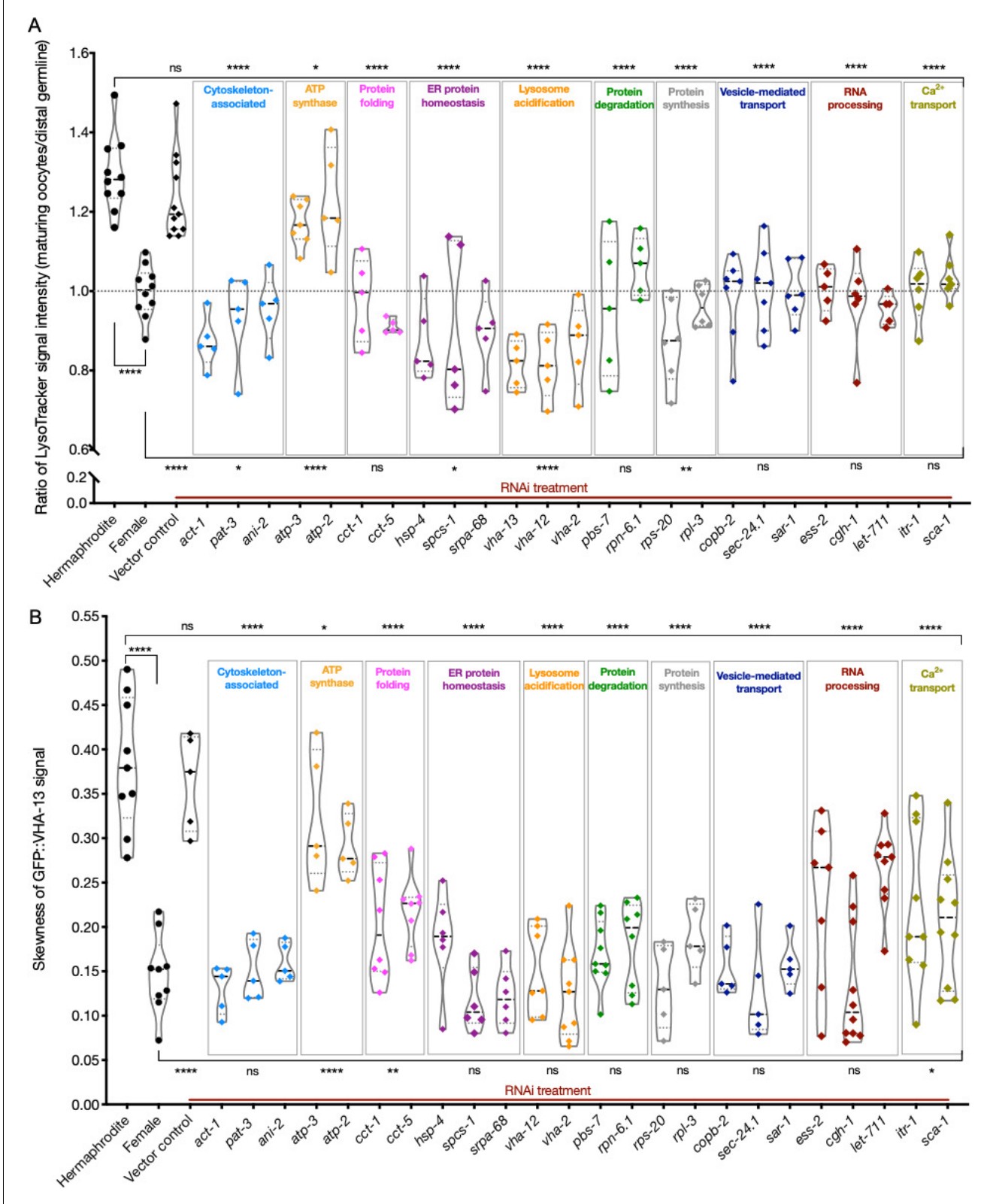

**Figure 4.** Lysosome acidification and GFP::VHA-13 localization following knockdown of genes identified by the screen. (**A**) LysoTracker staining in hermaphrodites (SA115) subjected to RNAi. The LysoTracker signal is represented as the ratio of proximal oocyte to distal germline staining. Staining ratios obtained from hermaphrodite (N2E) and female (CF4101) animals were used as controls (~1.3 and ~1.0, respectively). Each data point represents the average of values from three different locations in the most proximal oocyte from a single animal. The presence of GFP::RHO-1 aggregates in the

*Figure 4 continued on next page*

*Figure 4 continued*

same animals confirmed successful gene knockdown (not shown). (B) Analysis of GFP::VHA-13 localization, assessed by the degree of GFP skewness (inhomogeneity) in hermaphrodites (PHX1414) subjected to RNAi, and compared to control hermaphrodites and females (PHX1414 and CF4599). Each data point represents the average of 10 skewness values from different regions within the first three proximal oocytes of a single animal. The presence of aggregates in GFP::RHO-1 animals (SA115) treated in parallel was used as a control to verify successful gene knockdown (not shown). For (A, B), Mann–Whitney test was used to determine the statistical significance. The top and bottom rows of significance indicators in each experiment compare the mean values from each process category to hermaphrodite and female animals, respectively. p-Values: p>0.05 = n.s., *p≤0.05, **p≤0.01, and ****p≤0.0001.

The online version of this article includes the following source data and figure supplement(s) for figure 4:

**Source data 1.** LysoTracker signal intensity ratios (maturing oocytes/distal germline) in assay pool knockdowns.

**Source data 2.** Skewness of oocyte GFP::VHA-13 signal in assay pool knockdowns.

**Figure supplement 1.** Lysosomal acidification and GFP::VHA-13 localization in the germline following knockdowns of orphan genes.

transport (*itr-1, sca-1*) demonstrated intermediate values of skewness, indicative of at least a partial reduction of GFP::VHA-13 puncta (*Figure 4B*). Similarly, the orphan-gene inhibitions demonstrated reduced GFP skewness, with several candidates showing only intermediate effects (*kin-2, par-5, xpo-1, rmd-2, lgc-46, aco2, imb-1*) (*Figure 4—figure supplement 1B*). As anticipated, neither of the two ATP synthase gene knockdowns affected GFP::VHA-13 puncta formation (*Figure 4B*). Together, these data support the model that the failure of the gene knockdowns to acidify lysosomes is due, at least in part, to a failure to assemble the lysosomal V-ATPase.

## Two distinct, mysterious roles for mitochondria in protein aggregation

Mitochondria undergo a morphological and metabolic shift in response to signals from sperm, a shift that is required for aggregate removal (*Bohnert and Kenyon, 2017*). Unfertilized female oocytes exhibit a high mitochondrial membrane potential ($\Delta\Psi$) relative to hermaphrodite oocytes or to female oocytes exposed to sperm by mating. Loss of the mitochondrial ATP synthase, which discharges the proton gradient during the generation of ATP, elevates $\Delta\Psi$ in hermaphrodites and causes protein aggregates to accumulate (*Bohnert and Kenyon, 2017*). As expected, subunits from the mitochondrial ATP synthase complex (*atp-2, atp-3*) emerged from the screen. Previously, we found that lysosomal V-ATPase activity was required for the shift in mitochondrial membrane potential (*Bohnert and Kenyon, 2017*). To test the prediction that our screen hits, which all inhibited lysosome acidification, would also block the $\Delta\Psi$ switch, we stained hermaphrodites with the dye DiOC6 (3), whose signal is sensitive to $\Delta\Psi$. Again, to control for dye uptake, we measured the ratio of DiOC6(3) staining intensities between proximal oocytes and the distal syncytial germline. As predicted, all of the knockdowns prevented the $\Delta\Psi$ decrease in proximal hermaphrodite oocytes. Unexpectedly, in many cases the mitochondria exhibited $\Delta\Psi$ ratios even higher than those of females (*Figure 5A, Figure 5—figure supplement 1A*), suggesting that the female oocyte $\Delta\Psi$ is not the default state, but potentially an optimal point within a broader range of possibilities.

The observation that gene knockdowns from all the functional categories caused a relatively high oocyte $\Delta\Psi$ value prompted us to question whether a high mitochondrial membrane potential might be necessary for protein-aggregate accumulation. To test this, we asked whether exposing females expressing germline GFP::RHO-1 to conditions that lower $\Delta\Psi$, namely, to electron transport chain (ETC) RNAi knockdowns, might reduce protein aggregation. However, neither *cyc-1* nor *nuo-2* RNAi prevented females from accumulating aggregates, even though $\Delta\Psi$ was reduced (*Figure 5B, Figure 5—figure supplement 1B, C*). Thus, a high mitochondrial membrane potential is not a prerequisite for protein aggregation.

RNAi inhibition of ATP synthase has been shown to reduce ATP levels in worms, as does RNAi inhibition of electron transport (*Dillin et al., 2002; Lee et al., 2003*). To ask whether inhibiting ATP synthase was likely to cause protein aggregation in hermaphrodites by limiting ATP levels, we analyzed hermaphrodites subjected to ETC RNAi. The treatments reduced oocyte ATP levels (*Figure 5—figure supplement 2*) and reduced body size and growth rate (*Figure 5B*), two known correlates of reduced ATP levels. However, these perturbations did not cause protein aggregation in hermaphrodite oocytes (*Figure 5B*). This finding suggests the interesting hypothesis that loss of ATP synthase causes aggregates to accumulate in hermaphrodites not because it limits ATP levels, but for a different reason (see Discussion).

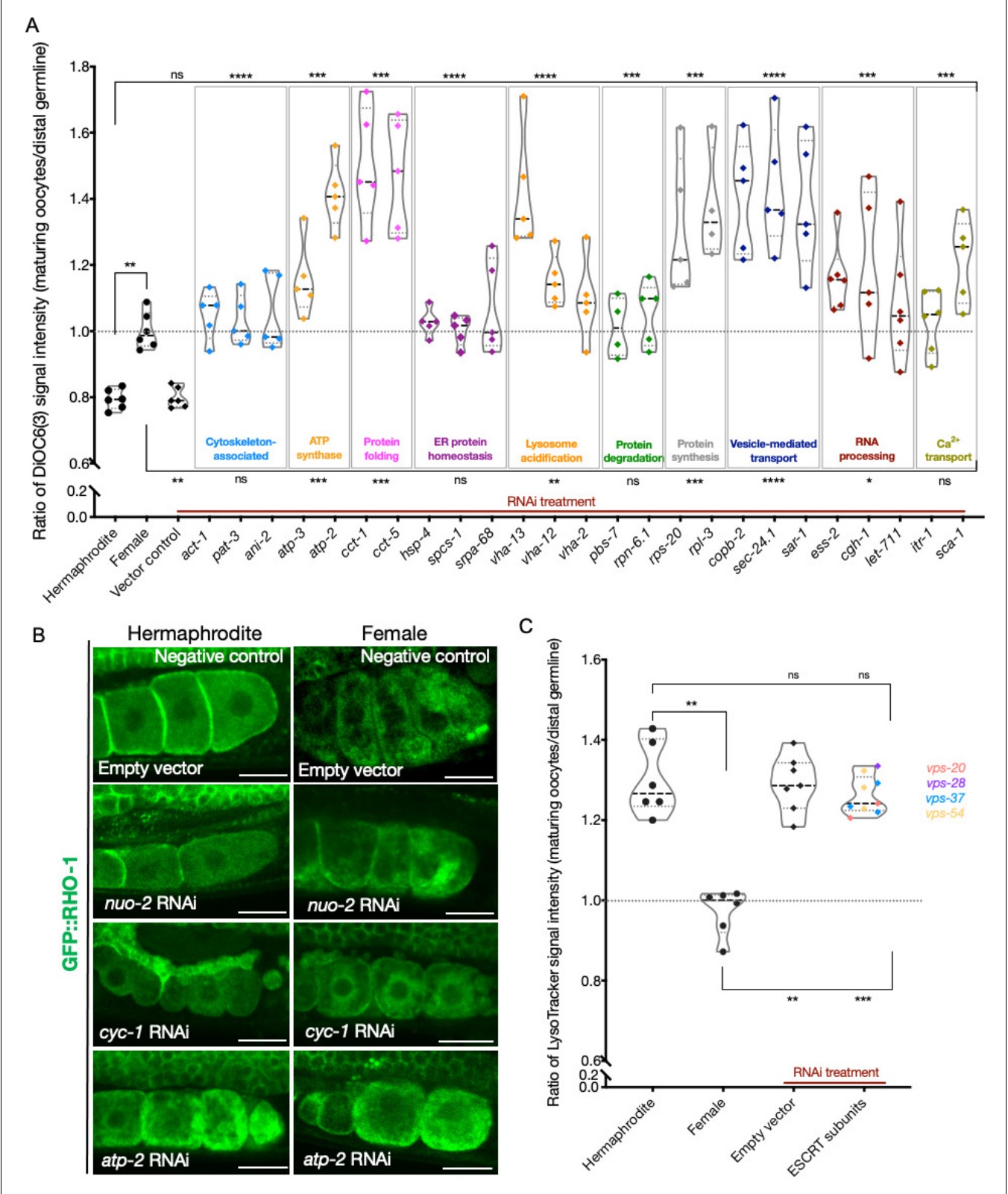

**Figure 5.** Analysis of mitochondrial membrane potential and ESCRT-complex subunits. (**A**) Analysis of oocyte ΔΨ using DiOC6(3) staining in RNAi-treated hermaphrodites (OCF15), represented as the ratio of proximal oocyte to distal germline staining. Staining ratios obtained from N2E hermaphrodites (~0.8) and CF4101 females (~1) were used as controls for comparison. Each data point represents the average of values from three different locations in the most proximal oocyte of a single animal. When applicable, alteration in endoplasmic reticulum (ER) morphology was used as a

*Figure 5 continued on next page*

*Figure 5 continued*

control to determine successful gene knockdown in the same animals. The presence of GFP::RHO-1 aggregates in SA115 animals treated in parallel was also used as an RNAi control (not shown). (**B**) Visualization of GFP::RHO-1 aggregates in oocytes of hermaphrodites (SA115) and females (CF4115) subjected to disruption of mitochondrial electron transport chain (ETC) genes. Successful ETC disruption was confirmed by smaller body sizes (visible in the images) and severely reduced brood sizes, similar to ATP synthase subunit knockdowns. Scale bars, 25 μm. (**C**) LysoTracker staining in hermaphrodites (SA115) subjected to ESCRT-complex RNAi, represented as the ratio of proximal oocyte to distal germline staining, and compared to hermaphrodites (N2E) and females (CF4101). Only animals exhibiting GFP::RHO-1 aggregates upon RNAi treatment (confirming knockdown; shown in *Figure 5—figure supplement 3A*) were included in the quantification. For (**A**, **C**), Mann–Whitney test was used to determine the statistical significance. The top and bottom rows of significance indicators in each experiment compare the mean values from each process category to hermaphrodite and female animals, respectively. p-Values: $p > 0.05$ = n.s., $*p \leq 0.05$, $**p \leq 0.01$, $***p \leq 0.001$, and $****p \leq 0.0001$.

The online version of this article includes the following source data and figure supplement(s) for figure 5:

**Source data 1.** DiOC6(3) signal intensity ratios (maturing oocytes/distal germline) in assay pool knockdowns.
**Source data 2.** LysoTracker signal intensity ratios (maturing oocytes/distal germline) in ESCRT subunit knockdowns.
**Source data 3.** BioTracker ATP-Red staining of oocytes.
**Source data 4.** Skewness of oocyte GFP::VHA-13 signal in ESCRT subunit knockdowns.
**Figure supplement 1.** Analysis of germline mitochondrial membrane potential; and effect of Krebs cycle gene knockdowns on GFP::RHO-1 aggregates.
**Figure supplement 2.** Analysis of germline ATP levels.
**Figure supplement 3.** Consequences of ESCRT gene knockdown on GFP::RHO-1 aggregates and GFP::VHA-13 localization.

Given the striking change in mitochondrial dynamics that takes place during oocyte maturation, we were surprised that in our screen we recovered only one additional gene encoding a mitochondrial-localized protein, *aco-2* (Krebs cycle aconitase). We tested several additional Krebs cycle gene knockdowns (*pdha-1*, *idhg-1*, *ogdh-1*, *sucl-1*, and *sdhd-1*) and found that α-ketoglutarate dehydrogenase (*ogdh-1*) and succinate dehydrogenase (*sdhd-1*) RNAi also led to oocyte GFP::RHO-1 aggregation (*Figure 5—figure supplement 1D*). Notably, unlike ATP synthase inhibition, *aco-2* inhibition prevented lysosome acidification (*Figure 4—figure supplement 1A*), indicating that the Krebs cycle and ATP synthase likely have distinct roles in the protein-aggregation pathway.

## ESCRT-complex genes are required for the clearance of protein aggregates

The direct mechanism by which protein aggregates are removed during oocyte maturation is not known with certainty, although morphologically, the process resembles lysosomal microautophagy (*Bohnert and Kenyon, 2017*). Microautophagy has not been described previously in *C. elegans*; however, orthologs of genes encoding components of the ESCRT machinery, which mediates microautophagy in yeast and mammals, are present (*Sahu et al., 2011*; *Williams and Urbé, 2007*). We identified two ESCRT subunits (*vps-37* and *vps-54*) in our primary screen, but they did not pass our initial validation cutoff (*Supplementary file 3*). To explore the possibility that these were false negatives, we retested these RNAi clones along with RNAi clones inhibiting additional ESCRT subunits (*tsg-101*, *vps-20*, and *vps-28*). In these experiments, the ESCRT subunits *vps-20*, *vps-28*, *vps-37*, and *vps-54* tested positive with subtle aggregation phenotypes (although not observed in all animals tested) (*Figure 5—figure supplement 3A*), supporting the model that aggregates are removed via microautophagy. Importantly, the degree of GFP::VHA-13 localization to puncta in these knockdowns was comparable to that of hermaphrodites (*Figure 5—figure supplement 3B, C*), and they also permitted acidification of oocyte lysosomes (*Figure 5C, Figure 5—figure supplement 3C*). Taken together, these findings suggest that the ESCRT machinery operates downstream of V-ATPase assembly and lysosome acidification to prevent aggregate accumulation.

## Mechanisms enhancing germline proteostasis also operate in the soma

Widespread proteostasis collapse is an inherent part of aging in *C. elegans* (*Ben-Zvi et al., 2009*; *David et al., 2010*; *Taylor and Dillin, 2011*; *Walther et al., 2017*), and mechanisms that extend lifespan invariably enhance cellular proteostasis. Understanding how the immortal germ lineage maintains cellular quality and proteostasis across the generations could suggest new ways to enhance proteostasis in the soma and delay organismal aging. Our lab and others have shown that many endogenous proteins become insoluble with age and can form visible aggregates when

overexpressed with fluorescent tags (*David et al., 2010*; *Huang et al., 2019*; *Reis-Rodrigues et al., 2012*; *Roux et al., 2016*). To visualize age-related protein aggregation, we monitored the casein kinase subunit KIN-19, which forms insoluble aggregates in an age-dependent manner in wild-type somatic tissues (*David et al., 2010*). To do this, we fluorescently tagged the endogenous *kin-19* gene specifically in the soma using split-wrmScarlet (*Goudeau et al., 2021*; *Supplementary file 2*). To avoid interference from the bright intestinal autofluorescence, we examined KIN-19::split-wrmScarlet in the head. The signal was mainly diffuse in young day 1 adults but began to exhibit a striking punctate appearance as early as day 4 of adulthood (*Figure 6A, B*, *Figure 6—figure supplement 1*). To the best of our knowledge, this tool is the first used to visualize age-dependent aggregation of an endogenously expressed protein in *C. elegans*.

To ask whether the germline proteostasis genes we identified in our screen might also affect somatic protein aggregation, we screened our germline aggregation 'assay pool' for gene knockdowns that caused KIN-19 to aggregate prematurely in head tissues. We found that multiple RNAi knockdowns that impaired lysosomal acidification in the germline-induced premature KIN-19::split-wrmScarlet aggregation in the soma, most notably RNAi of genes encoding lysosomal V-ATPase subunits (*vha-12*, *vha-13*, *vha-16*, and *vha-19*), but also genes that, in the germline, were required for V-ATPase assembly (the actin gene *act-5* and the vesicle-mediated ER-to-Golgi transport gene *copb-2*) (*Figure 6C, D*). We were not able to visualize lysosome acidification in the head using Lyso-Tracker, but these knockdowns did affect lysosome acidification in the intestine, where the RNAi treatment led to fewer discrete acidic lysosome-related organelles, thereby reducing the skewness of the LysoTracker signal (*Figure 6E*, *Figure 6—figure supplement 2*). We also recovered the proteasome subunits *pbs-7* and *rpn-6.1*, indicating a direct or indirect role for proteasome function to maintain KIN-19 solubility in the aging soma (*Figure 6C, D*).

We also tested a second somatic-aggregation reporter, the polyQ protein Q35::YFP expressed in body wall muscles (*Morley et al., 2002*). Similar to KIN-19::split-wrmScarlet, *vha-12* and *vha-19* knockdowns accelerated the timing of age-related Q35::YFP aggregation, as did *copb-2* and *rpn-6.1* RNAi (*Figure 6—figure supplement 3*). Interestingly, a few additional genes were identified that were not hits in the KIN-19::split-wrmScarlet screen. These included the cytosolic chaperones *cct-1* and *cct-5*, another trafficking component *rab-11.1*, and three orphans with diverse cellular function, *imb-1*, *itr-1*, and *let-711* (*Figure 6—figure supplement 3* and *Supplementary file 1*). Further, knocking down one of the four ESCRT-complex subunits we tested, *vps-37*, also led to Q35::YFP aggregation (*Figure 6—figure supplement 3*). In contrast, none of the ESCRT gene knockdowns accelerated the KIN-19::split-wrmScarlet somatic aggregation, even in animals exposed to RNAi from the first larval stage, despite a significant decrease in mRNA levels in the two subunits measured (*Figure 6—figure supplement 4*). Together, these findings suggest that lysosomal microautophagy may prevent the accumulation of Q35::YFP aggregates, but future studies will be required for certainty.

## Discussion

Sperm-derived signals that trigger oocyte maturation in *C. elegans* initiate a global change in oocyte structure and function, including the rapid removal of protein aggregates. A key step in aggregate removal is the acidification of lysosomes, which subsequently engulf the aggregates (*Bohnert and Kenyon, 2017*). In addition, mitochondria, which undergo morphological and metabolic changes in response to maturation signals, appear to play an important regulatory role as inhibition of mitochondrial ATP synthase prevents the downshift in mitochondrial membrane potential normally triggered by sperm and causes aggregates to form even when sperm are present (*Bohnert and Kenyon, 2017*). In this study, we identified a key role for a third organelle, the ER, in proteostasis renewal, and, using genetic and cell biological approaches, we deepened our understanding of all three organelles, the ER, lysosomes, and mitochondria, in oocyte protein quality control.

### Limits of our genetic screen

To identify additional genes and processes involved in protein-aggregate metabolism in oocytes, we carried out a genome-wide screen for RNA inhibitions that cause proteins to aggregate even in the presence of sperm, namely, in hermaphrodites. We expected to identify genes and processes that are directly involved in the sperm-dependent proteostasis switch, as well as genes that function

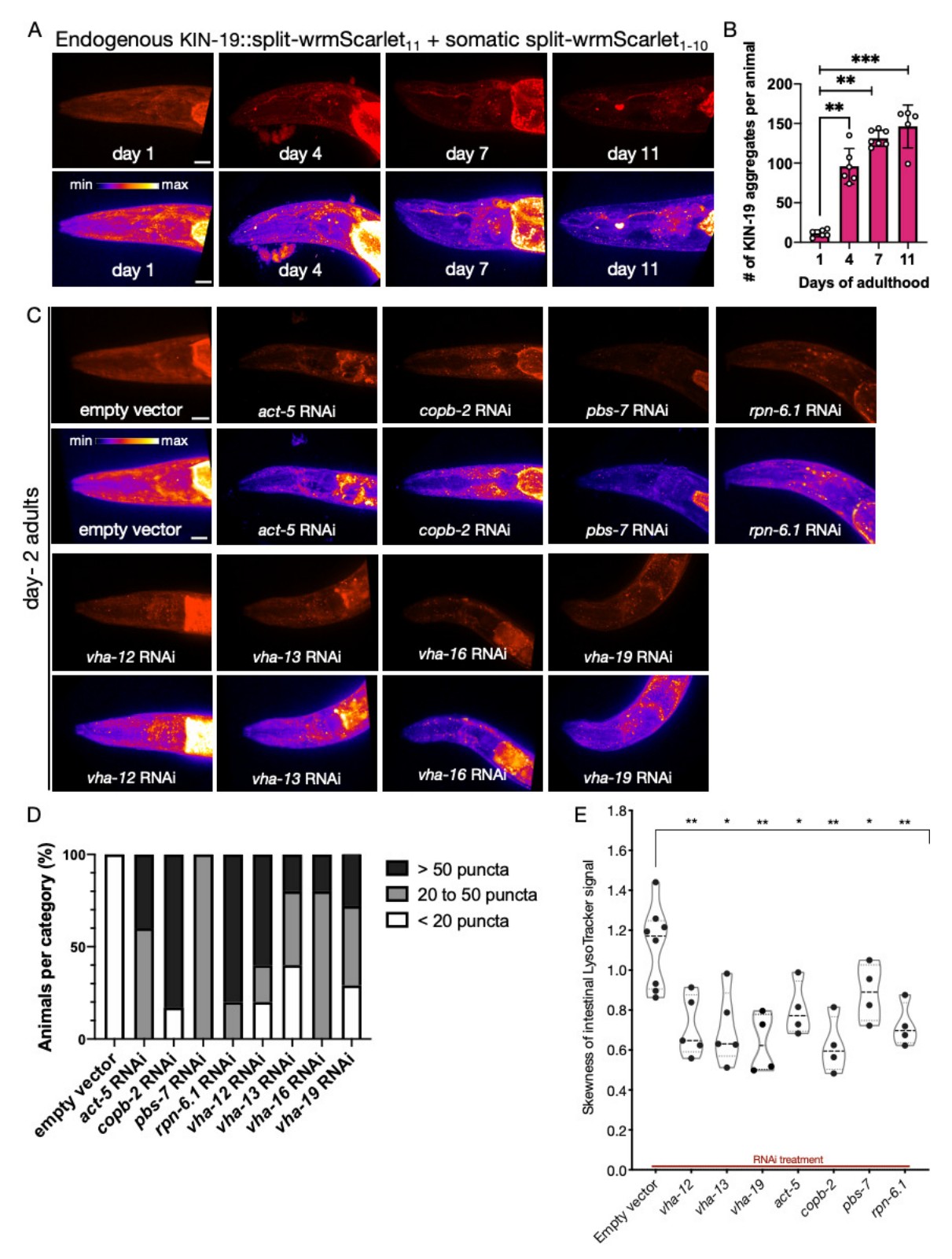

**Figure 6.** Analysis of protein aggregation and lysosome acidification in somatic tissues. (**A**) Representative images of CF4609 animals expressing KIN-19::split-wrmScarlet$_{11}$ and somatic split-wrmScarlet$_{1-10}$ at days 1, 4, 7, and 11 of adulthood. Maximum intensity projections of 3D stacks are shown in red (upper panels) and also using an intensity scale (lower panels). Scale bars, 10 µm. (**B**) Quantification of KIN-19::split-wrmScarlet aggregates per animal in the head region. The bright fluorescent signal in the pharyngeal region was not included in the quantification. Mann–Whitney test was used to

*Figure 6 continued on next page*

*Figure 6 continued*

determine the statistical significance. p-Values: \*\*p≤0.01 and \*\*\*p≤0.001. (**C**) Representative images of CF4609 animals expressing KIN-19::split-wrmScarlet$_{11}$ and somatic split-wrmScarlet$_{1-10}$ subjected to RNAi or empty-vector negative control, imaged for presence of KIN-19 aggregates at day 2 of adulthood. Scale bars, 10 µm. (**D**) Quantification of fluorescent KIN-19::split-wrmScarlet puncta (only from the head region) in CF4609 animals subjected to RNAi or empty-vector control. (**E**) Skewness of the intestinal LysoTracker signal in SA115 animals subjected to RNAi (shown in *Figure 6—figure supplement 2*). Each data point represents the average of six independent skewness values from the anterior intestinal region, near the pharyngeal bulb, in a single animal. The simultaneous presence of GFP::RHO-1 aggregates in oocytes was used as a control for successful gene knockdown in each animal (not shown). Mann–Whitney test was used to determine the statistical significance. p-Values: \*p≤0.05 and \*\*p≤0.01.

The online version of this article includes the following source data and figure supplement(s) for figure 6:

**Source data 1.** Number of KIN-19 aggregates per animal.
**Source data 2.** Skewness of intestinal LysoTracker signal.
**Source data 3.** Quantitative RT-PCR analysis of mRNA levels in ESCRT subunit knockdowns.
**Figure supplement 1.** Supplementary data related to *Figure 6B*.
**Figure supplement 2.** LysoTracker staining in the intestines of animals subjected to gene knockdowns that caused KIN-19::split-wrmScarlet aggregation in the head.
**Figure supplement 3.** Q35::YFP aggregation in body-wall muscle with age; and following assay-pool knockdowns.
**Figure supplement 4.** Investigating the contribution of ESCRT genes in preventing KIN-19 aggregation.

more indirectly in protein homeostasis and cellular quality control. Many of the genes we recovered affected large multiprotein complexes, such as the ribosome and proteasome, making a compelling argument for these complexes influencing protein aggregation, directly or indirectly. However, we may have missed processes carried out by only one or a few genes if the corresponding RNAi bacteria were missing from the RNAi library or otherwise problematic; or if their inhibition produced milder protein-aggregation phenotypes, like the ESCRT genes that failed our initial, stringent validation tests (*Supplementary file 3*).

To place the genes into a proteostasis pathway, we asked which hallmarks of the sperm-triggered aggregate-clearance pathway (lysosome acidification, mitochondrial membrane-potential shift, etc.) were blocked by the knockdowns (*Figure 7*). An intrinsic limitation of this strategy is that it only identifies the first role of a gene that acts in multiple steps in the process. For example, because inhibiting the proteasome prevents degradation of the proximal oocyte cell-fate determinant GLD-1 (*Bohnert and Kenyon, 2017*; *Kisielnicka et al., 2018*) and inhibiting various other chaperones prevents lysosome acidification, we cannot determine whether these proteins also play a direct role in the process of aggregate clearance itself. Finally, we note that perturbations that prevent sperm from producing the MSP oocyte maturation signal would also cause protein aggregation in hermaphrodites. The fact that the great majority of the knockdowns did not replicate female oocyte morphology exactly (*Figure 5A*, *Figure 5—figure supplement 1A*) argues against this possibility, but we have not ruled it out definitively.

## A role for the ER

Many of the genes we identified affected the ER (*Figure 7* and *Table 1*). All of these gene inhibitions prevented lysosome acidification, arguing that ER function influences aggregate formation indirectly, acting at a relatively early step in the proteostasis pathway (though ER function could be required again, at later stages, as well). Using TEM and fluorescent reporters, we validated and extended previous observations of ER reorganization during *C. elegans* oocyte maturation (*Langerak et al., 2019*). We found that the ER assumes a stacked arrangement, spatially distinct from GFP::RHO-1 protein aggregates, in the proximal oocytes of females. Importantly, we found that neither the ER stacking nor protein-aggregate accumulation were a consequence of long-term oocyte quiescence ('oocyte aging') as both were present as soon as female oocytes appeared during development. When oocyte maturation was triggered by mating, the ER underwent a rapid reorganization that preceded aggregate removal, consistent with a model that ER activity might be a precondition for aggregate clearance.

The ER-gene inhibitions fell into two classes. First was a set of genes with known ER functions that did not appear to affect ER morphology: *hsp-4* (BIP), *spcs-1* (signal peptidase complex subunit), and *srpa-68* (signal recognition particle). This finding argues that ER reorganization does not require ER function itself. The second class did affect ER morphology, generally causing the ER from

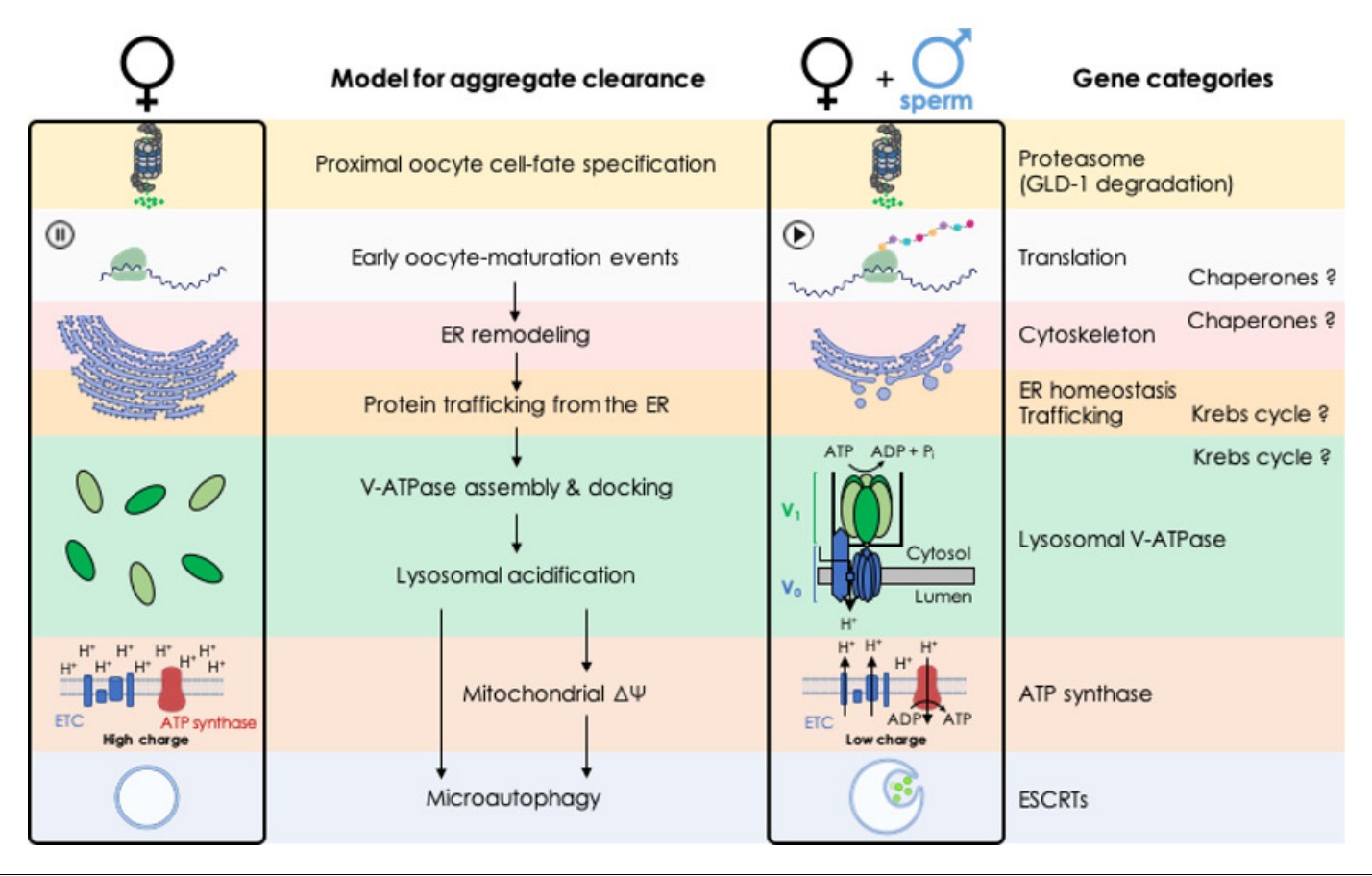

**Figure 7.** Model: biological processes implicated in the clearance of protein aggregates during oocyte maturation.

hermaphrodites to exhibit a bright patchy appearance characteristic of (though not necessarily identical to) that seen in female oocytes. Some of these gene knockdowns, such as that of *itr-1,* which encodes the channel that releases calcium stored in the ER, likely cause hermaphrodites to adopt a female-specific, stacked ER morphology. In *C. elegans*, the major wave of oocyte calcium release occurs after fertilization; this finding suggests that ER-dependent calcium metabolism plays an important earlier role as well. Other RNAi knockdowns may simply disrupt ER morphology more generally and might not identify genes that play an active role in the oocyte proteostasis switch. For example, the actin cytoskeleton is known to influence ER morphogenesis (*Poteryaev et al., 2005*), and we recovered a number of cytoskeletal gene hits. Notably, TRiC-complex chaperonin genes fell into this class as well. The TRiC-complex chaperonin can counteract protein aggregation directly; for example, by promoting correct protein folding during protein synthesis. However, TRiC is known to assist in folding cytoskeletal components including actin and tubulin (*Llorca et al., 2000*), and it was interesting to find that the TRiC-complex genes we recovered, *cct-1* and *cct-5,* acted indirectly by affecting ER morphology and lysosome acidification. As mentioned above, the proteasome too affects aggregate clearance indirectly. In general, it might be valuable to assay lysosome acidification when analyzing the functions of these chaperone/chaperonin systems on the accumulation of protein aggregates in other cellular systems to test for the possibility of indirect effects. That said, it is also possible that TRiC-complex and other chaperones act at multiple points in the proteostasis pathway, with both indirect and direct effects on protein aggregation.

Finally, we identified several genes that, to our knowledge, were not previously known to influence ER morphology; namely, *dlg-1*, *kin-2* and *ttr-14* (*Table 1*). How these genes act, directly or indirectly, to influence the ER could be interesting to explore in the future.

## Regulation of lysosome acidification

Lysosome acidification may be a conserved aspect of oocyte maturation as it takes place in Xenopus as well as *C. elegans* (*Bohnert and Kenyon, 2017*). A central goal of this study was to better understand how this lysosome acidification is regulated in worms. To this end, we first generated functional, fluorescently tagged V-ATPase subunits expressed from their own loci. However, for reasons that we do not understand, none of these proteins was visible in the germline. We then obtained a stable, single-copy MosSCI strain in which *gfp::vha-13* was expressed from a germline-specific promoter. In these animals, GFP::VHA-13 was visible, and we observed a striking sperm-dependent localization pattern. In the presence of sperm (in hermaphrodites), GFP::VHA-13 was located on lysosomes; that is, it assumed a punctate appearance that colocalized with LysoTracker signal. In the absence of sperm (in unmated females), the protein was distributed throughout the cytoplasm (*Figure 7*). The simplest interpretation of these findings is that signals from sperm trigger the localization of GFP::VHA-13 to lysosomes, and that this localization enables lysosome acidification. In principle, additional levels of control, such as translational repression of V-ATPase synthesis, could operate as well since our overexpressed construct could potentially override negative regulators, though we observed no evidence of this with serially diluted *gfp* RNAi treatments that progressively reduced visible GFP::VHA-13 protein levels.

In yeast, peripheral, cytoplasmic $V_1$ subunits of the V-ATPase (such as the VHA-13 ortholog VMA1) localize to lysosomes when the membrane-localized $V_0$ sector is trafficked there from the ER, where it is assembled (*Graham et al., 1998*). Thus, we propose that the extensive rearrangement in ER morphology triggered by sperm in *C. elegans* oocytes is coupled to movement of either preexisting or newly synthesized membrane-localized $V_0$ subunits of the V-ATPase from the ER to lysosomes, where they create docking sites for cytoplasmic, peripheral $V_1$ subunits. Consistent with this idea, RNAi of the ER genes, as well as each of the three genes required for vesicle trafficking from the ER to lysosomes, prevented GFP::VHA-13 localization to lysosomes, as did inhibiting genes encoding the membrane-localized V-ATPase $V_0$ subunits. Unfortunately, our efforts to observe the predicted sperm-activated movement of $V_0$ subunits from the ER to lysosomes were thwarted by our inability to visualize either of two functional wrmScarlet-tagged $V_0$ subunits, even when overexpressed using MosSCI technology. In the future, immuno-gold EM labeling of endogenous $V_0$ subunits might enable this model to be tested more directly.

We note that an interesting puzzle remains unexplained: If the multi-subunit lysosomal V-ATPase complexes are present in such low amounts in mature oocytes that they cannot be seen when we fluorescently tag the endogenous *vha-13* gene, then why do they appear when the tagged VHA-13 protein is overexpressed? How does the excess GFP::VHA-13 protein dock onto lysosomes? Perhaps the ER supplies lysosomes with alternative docking sites for unassembled V-ATPase subunits or perhaps the stoichiometry change due to the excess GFP::VHA-13 subunits triggers the production of entirely new V-ATPase complexes that incorporate the overexpressed GFP::VHA-13 subunits. This remains an unexplained but intriguing mystery (*Hughes and Gottschling, 2012*).

The gene knockdowns identified by the screen nearly all prevented lysosomal acidification, reinforcing the central role of acidified lysosomes in clearing oocyte protein aggregates (*Figure 7*). Moreover, none of the knockdowns that failed to acidify lysosomes exhibited a punctate, lysosomal GFP::VHA-13 pattern (*Table 1*). This finding suggests that V-ATPase activity may not only be necessary but also sufficient for lysosome acidification; however, this interpretation remains tentative since we may simply have failed to identify the relevant genes in our screen. Finally, several of the knockdowns from our screen did not appear to affect ER morphology (and have not been reported to affect ER function), but did prevent V-ATPase localization, for example, *srab-17*, a chemosensory gene of the serpentine receptor class ab, and *lgc-46*, involved in chloride ion transport (*Table 1*). How, directly or indirectly, these proteins might influence V-ATPase localization and assembly is an interesting question to pursue in the future.

## Two roles for mitochondria

Like the ER and lysosomes, mitochondria undergo morphological and functional changes during *C. elegans* oocyte maturation. In response to sperm, the mitochondria of proximal oocytes reduce their relatively high membrane potential and shift from a fragmented to a tubular structure. Inhibiting mitochondrial ATP synthase preserves this high membrane potential even when sperm are present

and causes aggregates to accumulate. Injecting ADP, whose levels would be expected to rise as maturing oocytes initiate protein synthesis and acidify lysosomes, reduces the mitochondrial membrane potential in female oocytes, in an ATP synthase-dependent fashion. Together, these findings suggested the model that in the absence of sperm, mitochondria are held in a quiescent, poised state that relaxes as metabolic activity and ATP synthesis commence (*Bohnert and Kenyon, 2017*).

Whether *C. elegans* oocyte maturation is powered primarily by glycolysis or respiration is unknown; however, in principle, the accumulation of protein aggregates caused by inhibiting mitochondrial ATP synthase could reflect ATP limitation. However, if this were the case, then inhibiting components of the ETC should also cause protein aggregates to accumulate in hermaphrodites since this too limits ATP levels. However, although these knockdowns reduced oocyte ATP levels, body size, growth rates, and brood size, this was not the case; ETC knockdowns had no effect on protein aggregation. This unexpected finding suggests that ATP synthase knockdown prevents aggregate clearance not because of changes in ATP levels, but instead either because a continuously high mitochondrial membrane potential acts like a checkpoint to inhibit aggregate clearance (*Figure 7*) or because of another, unknown role of ATP synthase in protein homeostasis.

Our screen revealed an additional requirement for mitochondria in protein homeostasis; specifically, for Krebs cycle activity. Inhibiting mitochondrial aconitase (*aco-2*) produced a different phenotype from inhibiting ATP synthase; namely, a failure to acidify lysosomes. Why might this be? Again, the explanation seems unlikely to involve energy limitation since inhibiting ETC components or ATP synthase did not produce this phenotype. One possibility is that inhibiting the Krebs cycle affects lysosome acidification by affecting redox potential, as several Krebs cycle enzymes affecting protein aggregation (*ogdh-1* and *sdhd-1*) encode dehydrogenases that produce the reducing agents NADH and $FADH_2$, and aconitase supplies the substrate for another Krebs cycle dehydrogenase. Alternatively, other metabolites generated via Krebs cycle intermediates could influence the events leading to lysosome acidification. It will be interesting to explore this mechanism in the future.

## Microautophagy

Previously, we observed lysosomes engulfing protein aggregates in a process reminiscent of microautophagy (*Bohnert and Kenyon, 2017*). Before these observations, microautophagy had not been described in *C. elegans.* In this study, using a candidate approach, we showed that inhibiting ESCRT-complex members, which are known to mediate microautophagy in other organisms, prevents aggregate accumulation in maturing *C. elegans* oocytes. Moreover, they influence protein aggregation independently of lysosomal acidification, consistent with their having a direct role. These findings strengthen the case that microautophagy is an active proteostasis mechanism in *C. elegans,* and likely the mechanism by which aggregates are cleared in response to oocyte maturation signals (*Figure 7*).

## Implications for somatic proteostasis and aging

Together, our findings all point towards lysosomal acidification being a key determinative event in oocyte protein-aggregate clearance. Notably, the screen hits that affected the lysosomal V-ATPase and COPII (ER-to-lysosomal vesicle transport) had a striking ability to accelerate age-dependent protein aggregation in somatic cells. Thus, the same mechanisms that enhance oocyte proteostasis in the immortal germline likely enhance somatic proteostasis as well. These findings may have implications for aging and longevity more broadly. In yeast, lysosomal acidity declines with age, and overexpressing the yeast ortholog of the *C. elegans* V-ATPase gene *vha-13* delays this decline and extends replicative lifespan (*Hughes and Gottschling, 2012*). In worms, lysosome acidification also declines with age, and this decline is delayed in long-lived mutants (*Baxi et al., 2017*; *Sun et al., 2020*). Whether increasing V-ATPase function might be sufficient to extend lifespan in *C. elegans* is unclear, but overproduction of the transcription factor HLH-30/TFEB, which augments many lysosomal functions, has been shown to do so (*Lapierre et al., 2013*; *O'Rourke and Ruvkun, 2013*), and in mammals, similar perturbations slow the aging of the immune system (*Zhang et al., 2019*).

In summary, a genetic screen for proteostasis enhancers in the immortal germ lineage has revealed a landscape of interconnected processes spanning protein translation, cytoskeletal dynamics, Krebs cycle activity, ER activation and vesicle transport, that together enable lysosome acidification and clearance of protein aggregates during oocyte maturation. Lysosomal acidification

maintains proteostasis in the aging soma as well. In addition to raising many new questions for future investigations, our findings support a growing body of evidence indicating that interventions which rejuvenate lysosomal activity during aging might yield new ways to increase human health and longevity.

# Materials and methods

**Key resources table**

| Reagent type (species) or resource | Designation | Source or reference | Identifiers | Additional information |
|---|---|---|---|---|
| RNAi reagents (*Caenorhabditis elegans*) | Multiple | *C. elegans* RNAi Collection (Source BioScience) | RRID:SCR_017064 | *Supplementary file 1* |
| Strains, (*Caenorhabditis elegans*) | Multiple | Caenorhabditis Genetics Center (CGC) | RRID:SCR_007341 | *Supplementary file 2* |
| Software, algorithm | GraphPad Prism 9.0.1 | GraphPad | RRID:SCR_002798 | |
| Software, algorithm | ImageJ/Fiji | Fiji | RRID:SCR_002285 | |

## *C. elegans* strains and maintenance

All strains used in this study are listed in *Supplementary file 2*. Hermaphrodites expressing germline transgenes were maintained at 25°C to delay silencing. Mutants carrying the thermosensitive, feminizing *fem-1(hc17ts)* allele, which causes cells that would normally become sperm instead to become oocytes, were maintained at the permissive temperature of 15°C and moved to 25°C for feminization prior to hatching, or at the L1 stage. Animals were grown under standard laboratory conditions (*Brenner, 1974*) and maintained on Nematode Growth Medium (NGM) seeded with OP50 bacteria, unless otherwise indicated.

## Generation of *C. elegans* strains

### Strain generation: genetic crosses

The following *C. elegans* strains were created by standard genetic crosses: CF4542 *fem-1(hc17ts) IV; ocfIs2[Ppie-1:mCherry::sp12::pie-1 3'UTR Cbr-unc-119(+)]*, CF4552 *tjIs1[Ppie-1::gfp::rho-1 Cbr-unc-119(+)]; ocfIs2 [Ppie-1:mCherry::sp12::pie-1 3'UTR Cbr-unc-119(+)]*, CF4557 *fem-1(hc17ts) IV; tjIs1 [Ppie-1::gfp::rho-1 Cbr-unc-119(+)]; ocfIs2 [Ppie-1:mCherry::sp12::pie-1 3'UTR Cbr-unc-119(+)]*, CF4559 *zuIs45[Pnmy-2::nmy-2::gfp Cbr-unc-119(+)] V; ocfIs2 [Ppie-1:mCherry::sp12::pie-1 3'UTR Cbr-unc-119(+)]*, CF4560 *fem-1(hc17ts) IV; zuIs45[Pnmy-2::nmy-2::gfp Cbr-unc-119(+)] V; ocfIs2 [Ppie-1:mCherry::sp12::pie-1 3'UTR Cbr-unc-119(+)]* and CF4599 *sybSi35[Ppie-1::gfp::vha-13::vha-13 3'UTR Cbr-unc-119(+)] II; unc-119(ed3/+) III; fem-1(hc17ts) IV.*

### Strain generation: Mos1-mediated single-copy insertion

The strains PHX1414 *sybSi35[Ppie-1::gfp::vha-13::vha-13 3'UTR Cbr-unc-119(+)] II; unc-119(ed3) III* and PHX2690 *sybIs2690[Ppie-1::vha-7::wrmScarlet::vha-7 3'UTR Cbr-unc-119(+)] I; unc-119(ed3) III* were generated by SunyBiotech using MosSCI services.

### Strain generation: CRISPR/Cas9-triggered homologous recombination

Split-wrmScarlet$_{11}$ was introduced at the C-terminus of KIN-19 using CRISPR/Cas9 and single-stranded oligodeoxynucleotide template (ssODN) in the strain CF4582 expressing split-wrmScarlet$_{1-10}$ in somatic tissues (driven by the *eft-3* promoter and *unc-54* 3'UTR) to obtain the strain CF4609. CRISPR insertion of split-wrmScarlet$_{11}$ was performed following published protocols (*Goudeau et al., 2021*; *Paix et al., 2016*; *Paix et al., 2015*). Briefly, ribonucleoprotein complexes (protein Cas9, tracrRNA, crRNA) and ssODN were microinjected into the gonads of young adults using standard methods (*Evans, 2006*). Injected worms were singled and placed at 25°C overnight. The crRNA and ssODN template sequences used to generate split-wrmScarlet$_{11}$ knock-in at the C

terminus of *kin-19* are the following: ssDNA_kin-19C: ACAGGGAGCTACCGTTCCATCAGCTGGAG TTCCAGCTGGAGTTGCACCAGGAGGAACTACTCCACAGGGAGGAGGATCCTACACCGTCG TCGAGCAATACGAGAAGTCCGTCGCCCGTCACTGCACCGGAGGATAAgacgttttgttgccgtcctgaggctttttaatccaaaaaagcccatgttaaatcatgtactatc; *kin-19_C_crRNA:* GGCAACAAAACGTCTTACTG (TGG).

KIN-19::split-wrmScarlet$_{11}$ integrants were identified by screening for fluorescent progeny in the progeny of injected worms, and the correct sequence insertion was confirmed by Sanger sequencing (Genewiz). Primers used for sequencing: kin-19_F: AGGCTCAACAATCGCAATCC and kin-19_R: TTTGAGTGCCCGACATTGGG.

Strains PHX731 *vha-13(syb731[wrmScarlet::vha-13]) V*, PHX1049 *vha-13(syb1049[gfp::vha-13]) V*, PHX1198 *ieSi65[sun-1p::TIR1::sun-1 3'UTR + Cbr-unc-119(+)] II; unc-119(ed3) III; vha-11(syb1198[vha-11::gfp::degron]/+) IV*, PHX1363 *vha-4(syb1363[wrmScarlet::vha-4]/+) II*, and PHX1317 *vha-7 (syb1317[vha-7::wrmScarlet]) IV* were generated by SunyBiotech using CRISPR services for the Kenyon lab.

## Nucleic acid reagents

Synthetic nucleic acids were purchased from Integrated DNA Technologies (IDT). For split-wrmScarlet$_{11}$ knock-in, a 200-mer HDR template was ordered in ssODN form (synthetic single-stranded oligodeoxynucleotide donors) from IDT.

## RNA interference

Unless otherwise indicated, RNAi was initiated in the early L4 larval stage prior to formation of oocytes but following much of development. One exception was *fem-1* RNAi, which was initiated at hatching to ensure feminization. RNAi bacterial feeding was initiated and maintained for 36–40 hr prior to imaging to allow sufficient time for knockdown.

## Genome-wide screen

The Ahringer *C. elegans* RNAi collection containing 18,225 bacterial clones, targeting approximately 87% of currently annotated *C. elegans* genes (Source BioScience), was used for screening. A workflow for screening in liquid (*Lehner et al., 2006*) was modified suitably for our needs. For a single 96-well plate, SA115 (GFP::RHO-1) hermaphrodites were grown on NGM plates seeded with thick OP50 lawns (from two medium-sized plates) for ~4 days to obtain a high density of healthy and gravid young adults.

## Day 1

Animals were subjected to standard bleaching and the eggs were washed thoroughly and then incubated in M9 buffer overnight at 20°C on a rotating mixer to obtain synchronized L1-arrested animals.

## Day 2

L1 animals were transferred to NGM plates spotted with HT115 bacteria and incubated 24–28 hr at 25°C to reach the early L4 stage. The RNAi library plate was aseptically replicated into deep well plates (96-well, square; Greiner Bio-One) containing 1 ml LB (with 25 µg/ml carbenicillin and 12.5 µg/ml tetracycline) per well. Positive (*vha-13* and *atp-2*) and negative (empty L4440 vector) controls were inoculated manually into blank wells if available or into wells on a separate control plate. The cultures were grown overnight at 37°C with shaking (185 rpm).

## Day 3

Fresh media (1 ml LB with 25 µg/ml carbenicillin) was added to the wells and incubated at 37°C, 155 rpm for 2 hr. Subsequently, 4 mM isopropyl β-d-1-thiogalactopyranoside (IPTG; Promega) was added to each RNAi well for induction of dsRNA synthesis, and plates were incubated for 3–4 hr at 30°C and 155 rpm. Synchronized L4 animals were washed off the HT115 plates with M9 and resuspended in NGM broth (with 25 µg/ml carbenicillin and 1 mM IPTG). This worm suspension (20 µl, ~10–15 animals) was added to each well of a Cellvis flat-bottom glass plate with high-performance #1.5 cover glass. The deep-well plate with dsRNA-induced RNAi bacteria was centrifuged, and the

cells were resuspended in the same NGM broth as the worms. 35 µl of each cell suspension was transferred into the corresponding well of the Cellvis plate already seeded with L4 animals for bacterial RNAi feeding. The feeding plate was incubated at 20°C, 155 rpm with adequate humidity in the shaker to prevent worms from settling or drying out.

### Day 5

After checking for the presence of adults with fully developed germlines, 25 mM sodium azide (Sigma-Aldrich) was added to each well and the plate was subjected directly to imaging on a spinning-disk confocal microscope. Typically, approximately six such 96-well plates were handled simultaneously in a given round of screening. The presence of aggregates in 50% or more animals for a specific RNAi condition was considered a hit.

*Notes*: The ability to automate image acquisition was complicated by a combination of low GFP::RHO-1 reporter expression and the close apposition of gonads with the intestine, an organ with significant autofluorescence. The density of animals and temperature of incubation determine developmental rate and were optimized by multiple pilot trials. Liquid handling was performed using Multidrop Combi from Thermo Scientific (initial inoculation of RNAi bacteria; day 2) and Agilent Bravo (resuspension and mixing of worms and induced bacteria; day 3). Following the validation rounds, the identities of the RNAi clones emerging from the screen were verified by sequencing (Quintara Biosciences). Investigators were blinded to the identity of RNAi clones, except controls, during screening.

### RNAi feeding on agar plates

Individual cultures were inoculated in tubes containing 1 ml LB (with 25 µg/ml carbenicillin and 12.5 µg/ml tetracycline) and incubated at 37°C with shaking (200 rpm). The cultures were diluted 1:10 in fresh LB containing only carbenicillin and allowed to grow for a further 2 hr, after which IPTG was added (4 mM) and tubes moved to a 30°C shaker for dsRNA induction (3–4 hr). Induced cultures were spotted on NGM plates containing 50 µg/ml carbenicillin and 2 mM IPTG, and the bacterial lawn was allowed to grow for 24–36 hr at 30°C, prior to introducing synchronized L4 animals.

### Somatic KIN-19::split-wrmScarlet and polyQ (Q35::YFP) protein-aggregation RNAi screens

KIN-19::split-wrmScarlet-expressing CF4609 animals were grown from a synchronized population of L1 animals on HT115 bacteria until they reached the L4 stage and then were transferred to plates seeded with RNAi bacteria until they reached day 2 of adulthood. Animals treated with ESCRT-complex subunits were exposed to RNAi clones from the L1 larval stage (*Figure 6—figure supplement 4*). The animals' heads were imaged by confocal microscopy. Z-stacks were analyzed using ImageJ (Fiji); image manipulations consisted of maximum intensity projections of the entire stack. The images of KIN-19::split-wrmScarlet puncta were quantified by the investigator who acquired the images (*Figure 6B*), as well as by three blinded investigators (*Figure 6—figure supplement 1*).

AM140 animals expressing the polyQ reporter Q35::YFP in body wall muscles were similarly synchronized and subjected to all assay pool RNAi starting at the L4 larval stage. The Q35::YFP reporter undergoes age-related aggregation starting on day 3 of adulthood; hence, the animals were imaged on day 2. Germline GFP::RHO-1 aggregation in SA115 animals, performed in parallel, was used as a control for successful gene knockdown. All positives from the initial screen were subjected to a second validation round. The initial screen was performed in liquid media in 96-well plates, and the validation was performed on regular NGM plates at 20°C.

## Evaluation of screen hits and selection of 'assay pool' genes

The initial whole-genome primary screen yielded 367 candidate genes, whose knockdown starting at the L4 larval stage resulted in GFP::RHO-1-aggregate accumulation within oocytes of young-adult hermaphrodites. False positives were eliminated by three independent rounds of validation, again using the GFP::RHO-1 strain, in which all candidates were subjected to the same workflow as the primary screen. The validation rounds were performed in a blinded fashion by two independent experimentalists and included the positive controls *vha-13* and *atp-2* RNAi (*Bohnert and Kenyon, 2017*). A candidate was considered to be positive if aggregation was observed in >50% animals in at least

two out of the three rounds of validation. This led to the confirmation of 88 candidates required for preventing GFP::RHO-1 aggregation in the hermaphrodite germline. Next, we randomly selected 55 candidates (ensuring representation of all the GO categories in *Figure 1D*, as well as each individual orphan candidate) and subjected them to orthogonal verification using the NMY-2::GFP reporter. If the representative candidates from a particular GO category tested positive in this round of screening, the entire category was considered confirmed. Finally, we discarded candidates that were (i) known to be required for male gonad development or spermatogenesis, (ii) non-specific RNAi targets, or (iii) known targets of the RHO-1 protein, thereby confirming 81 verified genes from the screen (*Figure 1C*). Notably, our analysis did not allow the identification of false negatives and we manually tested some genes that we thought might be involved in the pathway [e.g., TOR pathway genes (which did produce a clear positive signal) and ESCRT-complex subunits].

For selecting the 'assay pool' of genes used in all further phenotypic screens, we picked candidates from each GO category that were associated with consistent GFP::RHO-1 aggregation, with intermediate to strong phenotypes. In categories such as translation or protein degradation, we specifically included at least one candidate of small and large ribosomal subunits or structural and regulatory proteasomal subunits, respectively. Similarly, in the actin cytoskeletal category, we ensured that both structural and regulatory genes were included. No other specific criteria were used to avoid any bias in candidate selection. All confirmed orphans were included in the assay pool without any further selection.

## Fluorescence microscopy

Confocal fluorescence images were acquired on a Nikon Eclipse Ti CSU-X1 equipped with 405, 488, 561, and 640 nm lasers. Emission was collected through 455/50, 525/36, 605/70, or 700/75 nm filters on an Andor iXon Ultra EMCCD camera using the NIS-Elements software. Images were visualized using ImageJ Fiji. Maintenance and selection of transgenic worms expressing fluorescent markers were performed using a Leica M165 FC fluorescent stereo microscope equipped with a Sola SE-V light source. Photobleaching and wide-field experiments were performed using a Leica TIRF microscope equipped with an infinity scanner using 488 nm laser excitation. Emission was collected through a GFP-T filter cube on a Hamamatsu ORCA-Flash4 camera.

### Preparing worms for microscopy

Worms imaged in 96-well plates were immobilized by adding sodium azide to the liquid medium at a concentration of 25 mM (except when measuring mitochondrial $\Delta\Psi$, where the medium was carefully removed, and animals immobilized with levamisole). For all other experiments, animals were picked from plates and placed in a drop of 2 mM levamisole (Sigma) on a 4% agarose pad resting on a glass slide, and then secured with a coverslip.

### Dye staining

For experiments involving several different RNAi-knockdown conditions, RNAi feeding was set up similar to the screen workflow. The dyes LysoTracker Red DND-99 and DiOC6(3) ([2 µM] and Bio-Tracker ATP-Red [8 µM]) were added to the liquid NGM prior to resuspension of dsRNA-induced RNAi bacterial cells. After addition of synchronized L4 animals, feeding and dye uptake were allowed to proceed at 20°C for 36–40 hr. The 96-well feeding plates were protected from light during this time. The plate was centrifuged, bacterial suspensions were carefully washed off, and the animals were resuspended in fresh M9 buffer for imaging. For experiments involving fewer knockdown conditions or pairwise comparisons between hermaphrodites and females, the dye feeding was performed on NGM agar plates as described previously (*Bohnert and Kenyon, 2017*).

To analyze lysosome acidification in maturing oocytes, the ratio of signal intensities between proximal oocytes and the distal germline was calculated. In case of intestinal LysoTracker staining, signal intensities were not measured due to highly variable dye uptake and staining, and signal skewness (inhomogeneity) was measured instead. In both cases, multiple regions of oocytes or the anterior intestine were sampled for intensity or skewness values, and averaged for a single animal.

## Analysis of skewness

For semi-quantitatively analyzing the presence of GFP::VHA-13 puncta in oocytes and LysoTracker-stained puncta within the intestine, we measured skewness of the fluorescence signal. Skewness; that is, degree of symmetry or homogeneity of the signal, indicative of distinct lysosomal localization, was utilized as an alternative to counting GFP::VHA-13 puncta in individual oocytes or acidic LysoTracker-stained intestinal granules. Skewness was measured using the inbuilt function in Fiji (Analyze > Measure > Skewness). A skewness value of zero represents homogenous signal or lack of distinct puncta. For both GFP::VHA-13 and intestinal LysoTracker staining, skewness values were measured for multiple different regions and the mean value was reported for an individual animal.

## Imaging oocyte ATP levels using a FRET-based sensor

The *pie-1* promoter, the AT1.03 ATP-sensor (*Imamura et al., 2009*) containing two introns (codon optimized for *C. elegans*), and the *tbb-2* 3′UTR sequences were cloned into pUC57. Germline gene expression was achieved using a microinjection-based protocol with diluted transgenic DNA (*Kelly et al., 1997*). The *Ppie-1::AT1.03::tbb-2* 3′UTR construct (5 ng/µl) was co-injected with PvuII-digested genomic DNA fragments from *Escherichia coli* (100 ng/µl), and pRF4 (20 ng/µl) was used as a co-injection marker. Young N2E hermaphrodites were placed at 25°C after injection, and progeny were inspected for sensor expression 48 hr later. Larvae expressing the sensor were picked and transferred to RNAi feeding plates (clones targeting ETC genes) and allowed to grow until day 2 of adulthood before imaging.

Imaging was performed on a wide-field DMI-8 equipped with a spectra-x light source using 427/10 or 510/10 excitation filters and a ×40/0.85 objective. Emission was collected through 475/20 or 540/21 emission filters using an FF444/520/590 dichroic and pco.edge 4.2 camera. For analysis, FRET-channel images were divided by donor-channel images using ImageJ Fiji after subtracting a background estimated from non-fluorescent worms and masked using the fluorescence from the acceptor image.

## Mating experiments

An individual female animal was immobilized on an agarose pad using 2 mM levamisole, and the reporter of interest was imaged in the anterior gonad arm. Levamisole was washed off immediately, and the animal was introduced to an NGM plate containing young CB1490 male animals in a small spot of OP50 bacteria (five males:one female). The interactions were monitored using a benchtop stereo microscope, and when mating occurred, the female was transferred immediately to the agarose pad and imaged again, ensuring that the same oocytes were visualized before and after mating. Successful mating events were tracked by gonadal sheath contractions in response to sperm and subsequently confirmed by successful ovulation. Changes within individual oocytes upon mating were monitored up to 90 min at 10 min intervals. An identical procedure was performed for control females, except for exposure to males. Specific regions within proximal oocytes of control female animals were compared at all time points to determine the extent of signal intensity change due to photobleaching, with increasing time (*Figure 2—figure supplement 5*). In addition, regions of the distal germline were also monitored in mated animals (data not shown) to verify that the disappearance of the signal representing ER clusters and protein aggregates was not due to photobleaching.

## KIN-19::split-wrmScarlet aggregate accumulation in somatic tissues

A synchronized population of KIN-19::split-wrmScarlet-expressing CF4609 animals was grown on OP50 from L1 at 20°C. Worms were transferred every other day to freshly seeded NGM plates to remove progeny. Worms were imaged by confocal microscopy at days 1, 4, 7, and 11 of adulthood.

## Transmission electron microscopy

N2E hermaphrodites and *fem-1(hc17ts)* females were cultured for five generations at 15°C without starvation or contamination at any point. Animals from 12 small plates of each strain were bleached, and the synchronized eggs were collected and washed thoroughly in M9 and strained through 40 µm strainers to remove adult carcasses. Eggs were then resuspended in fresh M9 and incubated at 25°C overnight on a rotating mixer. Synchronized L1 animals were moved to NGM plates seeded with OP50 and incubated at 25°C (~50 animals per plate). Newly developed day 1 adults were

further processed according to standard *C. elegans* TEM techniques (*Hall et al., 2012*). Briefly, the animals were loaded into 100-µm-deep specimen carriers and high-pressure frozen in a Bal-tec HPM (approximately 20 worms per carrier). After freezing, the carriers were transferred to vials containing freeze substitution (FS) media (1% OsO4% and 0.1% uranyl acetate in acetone), while still submerged in liquid $N_2$. Frozen samples were slowly warmed to approximately −15°C while being agitated on a rotating platform (adapted from *McDonald and Webb, 2011*). Following FS, samples were infiltrated with epon resin/acetone series (25% resin [1 hr], 50% resin [1 hr], 75% resin [overnight], 100% resin [1 hr], 100% resin [1 hr], and the resin from last step replaced with fresh resin for embedding) and subjected to polymerization at 60°C for 48 hr. For staining, thin sections (70 nm thick) were cut on an ultramicrotome, picked up on Formvar-coated 50 mesh grids, and post-stained for 7 min in 2% uranyl acetate (aq), followed by 7 min in lead citrate. Images were acquired on a Tecnai 12 TEM.

## Quantitative real-time PCR

CF4609 animals treated with *vps-20* RNAi, *vps-54* RNAi or empty-vector control were recovered after imaging, collected in Eppendorf DNA LoBind tube with 10 µl of Cells-to-Ct lysis buffer supplemented with DNase I (Thermo Fisher Scientific) and placed at −80°C for 30 min. Tubes were thawed on ice and processed on a bead-beater for 30 s at 4°C. Worm lysates were incubated at room temperature for 10 min, and lysis was stopped by adding 2 µl of Stop Buffer from the Cells-to-Ct kit. Samples were then incubated at 95°C, 5 min. cDNAs were synthesized from RNA templates using a reverse transcription mix following the manufacturer's instructions. qPCR was carried out on an QuantStudio 6 Flex Real-Time PCR System using a SYBR green-based real-time kit (Kapa Biosystems). The RNA polymerase II subunit *ama-1* was used as the housekeeping control. The qPCR primers were as follows: *vps-20* [TCCGATCAGGATAATGCGATTT]/[AGCGCCTGGAATTACCAAA]; *vps-54* [TTCACTCTTCACAGGTGAGTTC]/[CCTGTGATCTACATGATTCTCTCTC]; *ama-1* [GACGAG TCCAACGTACTCTCCAAC]/[TACTTGGGGCTCGATGGGC].

## Data analysis and statistics

ImageJ Fiji was used to quantify and analyze attributes from image data, and images from a given experimental set were scaled and processed identically for comparison. Preparation of graphs and statistical analyses was performed using GraphPad Prism 8. Statistical tests used to determine significance and p-values are described in the figure legends of each experiment. Each violin plot representation includes the median and quartiles, and distributions in bar graphs represent mean ± standard deviation. Figures displaying numerical values obtained by analysis of microscopy images of dye-stained worms report the average of multiple independent values (technical replicates) from individual animals (biological replicates), each of which are represented as data points. Experiments reporting qualitative phenotypes from microscopy images were performed at least three times (independently) with 10–15 animals per condition in each individual experiment (except the primary screen that was done only once but validated in three independent experiments). For all other cases, sample sizes and replicates are included in the corresponding figure legends. No statistical methods were used to predetermine sample sizes. Investigators were blinded to identity of RNAi clones (except for controls) while imaging multiple knockdown conditions in 96-well plates.

## Acknowledgements

We thank colleagues in the Kenyon lab and at Calico Life Sciences for advice and discussion, and Daniel Gottschling and Calvin Jan for comments on the manuscript. Special thanks to Andrew Dillin (University of California, Berkeley) for sharing his transmission electron microscopy resources, and Peichuan Zhang for help with converting the RNAi collection format. We thank Robert Keyser for help with liquid handling robotics for the primary screen, and Kayley Hake for suggesting skewness as a way to capture puncta/aggregate formation in microscopy images. Peichuan Zhang and Rex Kerr helped to analyze somatic KIN-19::split-wrmScarlet aggregation. We thank Di Chen for sharing the DCL565 and DCL569 strains. Some *C. elegans* strains were provided by the CGC, which is funded by the NIH Office of Research Infrastructure Programs (P40 OD010440). The endogenously tagged V-ATPase and MosSCI *C. elegans* strains were generated by SunyBiotech, China. Some illustrations in *Figures 1* and *7* were created with BioRender.com. The study was supported by Calico

Life Sciences, DHH was supported by NIH OD010943, and MeS was supported by HHMI funding to A Dillin.

## Additional information

### Competing interests

Madhuja Samaddar: Madhuja Samaddar is affiliated with Calico Life Sciences LLC. The author has no financial interests to declare. Jérôme Goudeau: Jérôme Goudeau is affiliated with Calico Life Sciences LLC. The author has no financial interests to declare. K Adam Bohnert: K. Adam Bohnert was affiliated with Calico Life Sciences LLC. The author has no financial interests to declare. Maria Ingaramo: Maria Ingaramo is affiliated with Calico Life Sciences LLC. The author has no financial interests to declare. Cynthia Kenyon: Cynthia Kenyon is affiliated with Calico Life Sciences LLC. The author has no financial interests to declare. The other authors declare that no competing interests exist.

### Funding

| Funder | Grant reference number | Author |
| --- | --- | --- |
| Calico Life Sciences LLC | | Madhuja Samaddar<br>Jérôme Goudeau<br>K Adam Bohnert<br>Maria Ingaramo<br>Cynthia Kenyon |
| National Institutes of Health | OD010943 | David H Hall |
| Howard Hughes Medical Institute | | Melissa Sanchez |

The funders had no role in study design, data collection and interpretation, or the decision to submit the work for publication.

### Author contributions

Madhuja Samaddar, Conceptualization, Data curation, Formal analysis, Validation, Investigation, Visualization, Methodology, Writing - original draft, Writing - review and editing; Jérôme Goudeau, Conceptualization, Data curation, Formal analysis, Validation, Investigation, Visualization, Methodology, Writing - review and editing; Melissa Sanchez, Data curation, Investigation, Sample preparation and imaging for TEM experiments; David H Hall, Formal analysis, Methodology, TEM experimental design and data interpretation; K Adam Bohnert, Validation, Investigation; Maria Ingaramo, Formal analysis, Investigation, Methodology, Provided help and advice for fluorescence microscopy experiments, ATP sensor analysis and data analysis; Cynthia Kenyon, Conceptualization, Resources, Supervision, Writing - original draft, Project administration, Writing - review and editing

### Author ORCIDs

Madhuja Samaddar ⓘD https://orcid.org/0000-0001-5046-1604
Jérôme Goudeau ⓘD https://orcid.org/0000-0002-2483-1955
David H Hall ⓘD http://orcid.org/0000-0001-8459-9820
Cynthia Kenyon ⓘD https://orcid.org/0000-0003-3446-2636

### Decision letter and Author response

Decision letter https://doi.org/10.7554/eLife.62653.sa1
Author response https://doi.org/10.7554/eLife.62653.sa2

## Additional files

### Supplementary files

• Supplementary file 1. Germline proteostasis candidates captured in the screen. List of candidates obtained from primary screening and validation rounds with GFP::RHO-1. The phenotypic strengths

and fertility status are indicated for each. Candidates that were discarded on the basis of their known functions, possible off-target effects, or the failure to pass orthogonal verification with NMY-2::GFP are indicated accordingly.

• Supplementary file 2. List of *C. elegans* strains. Description of all strains used in the study, including the corresponding genotypes and sources.

• Supplementary file 3. Candidates from primary screen that did not pass validation. This list is included because some of these candidates could be false negatives, as was the case for the ESCRT-complex genes. Also included are the gene ontology terms (biological process) they represent.

• Transparent reporting form

### Data availability

All data generated or analysed during this study are included in the manuscript and supporting files.

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
