## [Decision Letter]

**Acceptance summary:**

The revised manuscript details an elegant and well-designed screening approach and subsequent careful validation of the hits generated by the central screen. The result is an unbiased, near full genome and well-validated assessment of genes critical for clearing protein aggregates in the germline. The genes and mechanisms identified as promoting enhanced proteostasis in the immortal germ line likely play an important role in controlling lysosomal activity during organismal aging, opening up exciting avenues for future research into factors controlling healthy lifespan.

**Decision letter after peer review:**

Thank you for submitting your article "A genetic screen identifies new steps in oocyte maturation that enhance proteostasis in the immortal germ lineage" for consideration by *eLife*. Your article has been reviewed by 3 peer reviewers, one of whom is a member of our Board of Reviewing Editors, and the evaluation has been overseen by Jessica Tyler as the Senior Editor. The reviewers have opted to remain anonymous.

The reviewers have discussed the reviews with one another and the Reviewing Editor has drafted this decision to help you prepare a revised submission.

Summary:

The reviewers agreed that the manuscript reports a significant, technically strong data on genes involved in triggering the clearance of protein aggregates from mature oocytes that is mediated by the lysosome switch following fertilization.

The central genome-wide screen is enabled by an elegant and carefully designed approach, followed by carefully validation of hits using a staged experimental design and this results in a high quality dataset of genes involved. The reviewers also agreed that the evidence provided is generally strong and found the core conclusions convincing. There was some difference in opinion regarding the novelty of these insights, mainly due to most of the target genes falling into gene families and pathways previously implicated in this process. However, it was noted that there are also a number a novel and potentially important observations, including the identification of ESCRT-complex members. The screen provides an unbiased, near full genome, assessment of genes that are critical for clearing protein aggregates in oocytes. Because of this, I feel that the absence of hits to unexpected genes and pathways is also a relevant insight. The majority view was that on the strength of the evidence there is sufficient novelty and interest to accept the manuscript for publication in *eLife*, subject to adequate responses to several major comments. These comments are listed below.

Essential revisions for this paper:

1) Defects of clearance of protein aggregates from the mature oocytes upon RNAi treatment may also be due to defects/limitations in sperm signalling. An important control that is missing is to show that the effects derive from the oocytes physiology rather than from the sperm. This could be tested by mating RNAi-treated animals (for all the RNAis that passed the screen) with wild-type males, and following oocyte proteostasis. This can be done for a representative gene from each category (proton pump, mitochondria, er, microautophagy). If collection of such data is not feasible due to COVID related limitations, this concern should be discussed and the question addressed in follow up work.

2) The authors also suggest that proteostasis promoting mechanisms in the germline also operate in the soma. This statement is based on one somatic reporter whose aggregation was increased upon RNAi treatments that were shown to affect lysosome acidification. Given the central role of the lysosome in other proteostasis promoting pathways such as autophagy, the correlation to germline proteostasis could be coincidental. These concerns should be addressed either experimentally or the section should be toned done or removed (it does not affect the main conclusions of the paper).

3) Related to the same observation that knockdown of several of the genes identified also appears to affect protein aggregation in the soma (Figure 6) – and with reference to the final discussion, where the authors link their insights to possible future interventions to extend healthy lifespan, it would be informative to report healthspan or lifespan of worms subjected to those RNAi construct that increase endogenous protein aggregation (Figure 6D). Have these experiments not been done? Could some of these data be added? If not, this should be considered for follow up work as well. Failing to provide such data the question should be discussed.

Revisions expected in follow-up work:

1) The authors use reduced size and growth rate as proxy for reduced energy availability and ATP levels with ETC RNAi. One concern with this is that these data are not actually shown – making the consistency of this effect hard to evaluate. However, more importantly, if growth is reduced and animals are smaller, how do we know that ATP levels (esp. in oocytes) are in fact lower? With less energy used for growth, might the germline not be protected from this reduction in ATP levels? This should be addressed at least in writing and/or may be addressed experimentally in the follow up.

---

## [Author Response]

Essential revisions for this paper:1) Defects of clearance of protein aggregates from the mature oocytes upon RNAi treatment may also be due to defects/limitations in sperm signalling. An important control that is missing is to show that the effects derive from the oocytes physiology rather than from the sperm. This could be tested by mating RNAi-treated animals (for all the RNAis that passed the screen) with wild-type males, and following oocyte proteostasis. This can be done for a representative gene from each category (proton pump, mitochondria, er, microautophagy). If collection of such data is not feasible due to COVID related limitations, this concern should be discussed and the question addressed in follow up work.

Thank you very much for this comment. We agree, it should be addressed in the paper. The mating experiments are actually quite complicated and time consuming, as the mating events need to be observed in real time, and then the animals need to be transferred immediately to the confocal microscope to assay aggregate abundance, prior to ovulation. Because most of the RNAi treatments that produce aggregates disrupt fundamental aspects of oocyte cell biology (translation, lysosomal function, proteosome function, ER function, etc.), this careful timing, using young females, seems especially important, since we would want to ensure that the RNAi-treated oocytes are capable of aggregate clearance if their aggregation defect is due to defective sperm-signal (MSP) production. We did try another approach, using an auxin-inducible strategy we developed to clear a component of the MSP (sperm signal) response pathway to mimic mating, but unfortunately this strategy proved to be unfeasible. Thus, at this point we can say with certainty that the RNAi knockdowns do not only affect the sperm, as hermaphrodite oocytes exposed to our RNAi hits exhibit morphologies distinct from those seen in the stacked oocytes of normal females. Likewise, physiological aspects of oocytes from RNAi-treated hermaphrodites are distinct from females. For example, as we describe in the text, the ratio of the mitochondrial membrane potential (maturing oocytes / distal germline) is higher in the germlines of many RNAi-treated hermaphrodite than it is in normal females, as measured using DiOC6(3) (Figures 5A and Figure 5—figure supplement 1). Nonetheless, we cannot exclude the possibility that in some cases, gene knockdown causes aggregates to accumulate because it blocks MSP production by sperm, not because of the additional effects it had on oocytes. We now state this caveat in the section “limits of our genetic screen” (lines 524-528).

2) The authors also suggest that proteostasis promoting mechanisms in the germline also operate in the soma. This statement is based on one somatic reporter whose aggregation was increased upon RNAi treatments that were shown to affect lysosome acidification. Given the central role of the lysosome in other proteostasis promoting pathways such as autophagy, the correlation to germline proteostasis could be coincidental. These concerns should be addressed either experimentally or the section should be toned down or removed (it does not affect the main conclusions of the paper).

Thank you for this comment. We have now analyzed an additional somatic protein, the aggregation-prone polyQ protein, Q35, expressed in the body-wall muscle (Morley J et al., 2002). Similar to our results with the KIN-19 reporter, we find that a number of our germline hits lead to premature aggregation of the Q35::YFP reporter when knocked down using RNAi. While multiple genes identified in the Q35 screen overlap with those identified using KIN-19, there are also some interesting differences, some of which may reflect specific properties of different types of protein aggregates. For instance, *vps-37* RNAi resulted in aggregation of the Q35 reporter but not of KIN-19 (Figure 6—figure supplement 3 and 4). Notably, genes involved in macroautophagy (*bec-1, lgg-1, atg-9*) didn’t lead to aggregation in either somatic reporter under the conditions tested and criteria used for the rest of our analyses. Thus, together our results suggest that lysosome acidification is key for preventing protein aggregation in the *C. elegans* germline and soma, and likely involves mechanism(s) other than macroautophagy. We have updated the manuscript to include the new data in the Results (lines 471-484), Figures (Figure 6—figure supplement 3 and 4), and Materials and methods (lines 875-876 and 1032-1045).

3) Related to the same observation that knockdown of several of the genes identified also appears to affect protein aggregation in the soma (Figure 6) – and with reference to the final discussion, where the authors link their insights to possible future interventions to extend healthy lifespan, it would be informative to report healthspan or lifespan of worms subjected to those RNAi construct that increase endogenous protein aggregation (Figure 6D). Have these experiments not been done? Could some of these data be added? If not, this should be considered for follow up work as well. Failing to provide such data the question should be discussed.

Thank you for this comment. While we do not provide additional data regarding the healthspan or lifespan of animals subjected to RNAi that increased KIN-19 aggregation, most of the corresponding gene knockdowns have previously been reported to reduce lifespan. For example, the V-ATPase knockdown *vha-13(RNAi)* drastically shortens the lifespan of animals exposed to RNAi from L4 stage (Extended data figure 3 from Bohnert and Kenyon, 2017); RNAi treatments of most proteasome subunits during adulthood, including *pbs-7* RNAi and *rpn-6* RNAi, elicited a dramatic shortening of lifespan in wild-type animals (SI Table 1 from Ghazi et al., 2007). Likewise, *copb-2* RNAi (*F38E11.5* RNAi) shortens lifespan (Table S3 from Tacutu et al., 2012).

Revisions expected in follow-up work:1) The authors use reduced size and growth rate as proxy for reduced energy availability and ATP levels with ETC RNAi. One concern with this is that these data are not actually shown – making the consistency of this effect hard to evaluate. However, more importantly, if growth is reduced and animals are smaller, how do we know that ATP levels (esp. in oocytes) are in fact lower? With less energy used for growth, might the germline not be protected from this reduction in ATP levels? This should be addressed at least in writing and/or may be addressed experimentally in the follow up.

Although we did not provide specific quantification for the reduced size and growth rate in animals subjected to ETC knockdowns, the reduced sizes relative to the empty-vector-treated animals are visible in Figure 5B of the manuscript (both hermaphrodites and females); note scale bars. The effects of various ETC knockdowns on ATP levels and growth and development have also been reported previously by multiple groups. For example, reduced ATP levels were described in multiple respiratory knockdowns including *nuo-2, cyc-1* and *atp-3,* by Dillin A et al., 2002. Further, reduced growth and slow development have previously been reported for ETC knockdowns e.g., Melo and Ruvkun, 2012 (*atp-2*), Fraser AG et al., 2000 (*nuo-2*) and Zipperlen P et al., 2001 (*cyc-1*).

To specifically investigate the effect of ETC knockdowns on germline ATP levels, requested by the reviewer, we utilized two approaches. First, we fed the worms Biotracker ATPRed dye, a live-cell imaging dye, and we observed reduced oocyte ATP levels in ETC knockdowns relative to the control animals. In addition, we injected a modified germline-specific FRET-based ATP sensor, Ateam (Imamura et al., 2009), to probe oocyte ATP levels. Unfortunately, Ateam expression was found to be highly variable and only a small number of animals demonstrated sufficient expression for conducting the FRET experiments reliably. However, even with the limited numbers of animals, we observed a marked reduction in ATP levels in oocytes of animals subjected to ETC RNAi. Thus, the results from both of these approaches confirm that ETC gene knockdowns markedly reduce germline ATP levels, and that this tissue is not protected from ATP depletion. We have updated the manuscript to include the new information in the Results (line 398), Discussion (lines 650-651), Figures (Figure 5—figure supplement 2) and Materials and methods (lines 941 and 969-985).

References:

Bohnert KA, Kenyon C. 2017. A lysosomal switch triggers proteostasis renewal in the immortal *C. elegans* germ lineage. *Nature* 551:629–633. doi:10.1038/nature24620

David DC, Ollikainen N, Trinidad JC, Cary MP, Burlingame AL, Kenyon C. 2010. Widespread protein aggregation as an inherent part of aging in *C. elegans*. *PLoS Biol* 8:47–48.

doi:10.1371/journal.pbio.1000450

Dillin A, Hsu AL, Arantes-Oliveira N, Lehrer-Graiwer J, Hsin H, Fraser AG, Kamath RS, Ahringer J, Kenyon C. 2002. Rates of behavior and aging specified by mitochondrial function during development. *Science* doi:10.1126/science.1077780

Fraser, A., Kamath, R., Zipperlen, P. et al. Functional genomic analysis of *C. elegans* chromosome I by systematic RNA interference. Nature 408, 325–330 (2000).

https://doi.org/10.1038/35042517

Ghazi, A., Henis-Korenblit, S. and Kenyon, C. Regulation of *Caenorhabditis elegans* lifespan by a proteasomal E3 ligase complex. *Proc Natl Acad Sci U S A* 104, 5947–5952 (2007). doi:

10.1073/pnas.2012370118

Imamura, H., Huynh Nhat, K., Toagawa, H., Saito, K., et al. 2009. Visualization of ATP levels inside single living cells with fluorescence resonance energy transfer-based genetically encoded indicators. *Proc Natl Acad Sci U S A* 106 (37) 15651-56. doi: 10.1073/pnas.0904764106

MacQueen, A. J. *et al.* ACT-5 is an essential *Caenorhabditis elegans* actin required for intestinal microvilli formation. *Mol. Biol. Cell* 16, 3247–3259 (2005). doi: 10.1091/mbc.E04-12-1061

Melo, J., and Ruvkun, G., 2012. Inactivation of Conserved *C. elegans* Genes Engages Pathogen- and Xenobiotic-Associated Defenses. Cell 149: 452-466. doi:

10.1016/j.cell.2012.02.050

Morley, J., Brignull, H., Weyers, J., Morimoto, R. 2002. The threshold for polyglutamineexpansion protein aggregation and cellular toxicity is dynamic and influenced by aging in *Caenorhabditis elegans*. *Proc Natl Acad Sci U S A* 99: 10417–10422. doi:

10.1073/pnas.152161099

Tacutu, R. *et al.* Prediction of *C. elegans* longevity genes by human and worm longevity networks. *PLoS ONE* 7, e48282 (2012). doi: 10.1371/journal.pone.0048282

Zipperlen, P., Fraser, A., Kamath, R., Martinez-Campos, M., Ahringer, J. 2001. Roles for 147 embryonic lethal genes on *C. elegans* chromosome I identified by RNA interference and video microscopy. *EMBO J*. 20:3984-3992 doi: 10.1093/emboj/20.15.3984